# DisjunctiveNet: Neural Symbolic Learning via Differentiable Convexified Optimization Layers

**Shraman Pal** [* 1]   **Can Li** [* 1]

## Abstract

Many learning tasks in science and engineering are characterized by sparse datasets, which limits the effectiveness of purely data-driven approaches. At the same time, these problems are often accompanied by rich domain knowledge derived from physical laws, operational requirements, and expert heuristics. Such knowledge is frequently expressed as rules involving logical propositions and linear inequalities. Existing neuro-symbolic methods typically enforce these rules approximately through soft penalties, assume input-independent rules when designing specialized architectures, or rely on non-differentiable post-processing at inference time to achieve hard constraint satisfaction. While recent advances in differentiable optimization layers enable end-to-end feasibility enforcement within neural networks, extending these approaches to logical or mixed-integer rules remains challenging due to inherent nonconvexity. In this work, we propose a unified end-to-end framework for enforcing hard, input-dependent mixed integer linear constraints within neural networks. Our approach represents rules as disjunctive constraints and applies hierarchical convex relaxations to obtain convex hull formulations. These relaxations yield tractable linear constraints that can be embedded as differentiable optimization layers while enabling exact rule satisfaction. We demonstrate the effectiveness of the proposed framework on real-world datasets, achieving perfect rule satisfaction and strong predictive performance.

## 1. Introduction

Deep learning has achieved remarkable performance in applications such as natural language processing (Vaswani et al., 2023; Devlin et al., 2019), computer vision (He et al., 2016). Similar learning-based approaches are increasingly influential in scientific and engineering workflows, where models must capture complex physical processes (Karniadakis et al., 2021; Chen et al., 2018) or high-dimensional biological structure (Jumper et al., 2021). However, many real-world scientific and engineering applications operate in regimes with limited labeled data and frequent distribution shifts. In these settings, purely data-driven deep learning models can make predictions that violate known structural, physical, or safety constraints, undermining the reliability of such systems. At the same time, these domains often provide rich prior knowledge-physical laws, experimental protocols, and expert heuristics, that generally remain valid across such distributional shifts.

A broad literature on neuro-symbolic learning incorporates domain knowledge via *soft* penalties or differentiable surrogates for logic rules (Hu et al., 2020; Xu et al., 2018; Wang et al., 2019; Manhaeve et al., 2018; Giunchiglia et al., 2022). While effective in practice, soft enforcement does not guarantee feasibility and penalty coefficients can be costly to tune. In parallel, differentiable optimization layers provide *hard* enforcement for many continuous and convex constraint families (Amos & Kolter, 2017; Agrawal et al., 2019). Yet extending these layers to *logical* rules is challenging: operators such as implication and disjunction naturally induce nonconvex, disjoint feasible regions, and many scientific rules are *input-dependent* ("if-then"), activating only in specific regimes.

This paper introduces a framework for enforcing *input-dependent logical rules* over neural network (NN) outputs, where each rule is represented as a finite disjunction of linear constraints, corresponding to unions of polyhedra. We show that this constraint representation is as expressive as mixed-integer linear programming (MILP) constraints used by the optimization community and the Quantifier-Free Linear Real Arithmetic (QF-LRA) used by the neural symbolic community. To guarantee exact constraint satisfaction while retaining end-to-end differentiability, we adopt the

---

[1]Davidson School of Chemical Engineering, Purdue University, West Lafayette, USA. Correspondence to: Shraman Pal <pal52@purdue.edu>, Can Li <canli@purdue.edu>.

*Proceedings of the 43rd International Conference on Machine Learning*, Seoul, South Korea. PMLR 306, 2026. Copyright 2026 by the author(s).

basic step hierachy from the disjunctive programming literature (Balas, 2018) to sequentially convexify the disjunctive constraints, from a conjunctive normal form (CNF) to a disjunctive normal form (DNF). We show that the convex hull reformulation of the DNF admits a differentiable linear programming (LP) representation while guaranteeing satisfaction of the original logical constraints. We evaluate the proposed approach on synthetic control tasks and a real-world single-cell RNA sequencing (scRNA-seq) classification benchmark, where domain knowledge naturally manifests as input-dependent logical rules. Across all settings, enforcing logical structure provides a strong inductive bias in data-scarce regimes and substantially improves rule satisfaction at inference time.

## 2. Related Work

**Continuous constraints**   For continuous constraints, most learning-based approaches fall into soft or hard enforcement categories. Soft methods encourage feasibility through penalty or augmented Lagrangian objectives and primal-dual style updates (Fioretto et al., 2020; Park & Van Hentenryck, 2023), as well as differentiable correction schemes such as unrolled gradient iterations or constraint-based completion (Donti et al., 2021). While often effective empirically, these approaches generally do not provide feasibility guarantees at either training or inference time. In contrast, hard methods enforce feasibility by design, for example, by embedding an optimization layer in the forward pass and differentiating through Karush-Kuhn-Tucker (KKT) conditions using implicit differentiation (Amos & Kolter, 2017; Agrawal et al., 2019). Beyond optimization layers, recent work has proposed tailored projection-based methods relying on closed-form solutions for special cases like linear equalities (Chen et al., 2024), gauge functions for convex constraints(Constante-Flores et al., 2025; Tabas & Zhang, 2022; Liang et al., 2024; Tordesillas et al., 2023), inner approximations of the feasible region (Frerix et al., 2020; Zheng et al., 2021), and iterative projection procedures (Lastrucci & Schweidtmann, 2025; Iftakher et al., 2025; Nguyen & Donti, 2025). Despite these advances to enforce hard constraints, existing approaches are largely restricted to continuous, and in particular convex constraints, and do not directly address logical or discrete constraints.

**Logical constraints**   An orthogonal line of work has been developed by the neuro-symbolic AI community (Giunchiglia et al., 2022), which incorporates logical or symbolic rules into NNs. A common approach is to modify the loss function, where logical constraints are translated into differentiable penalty terms that bias training toward rule satisfaction (Xu et al., 2018; Hu et al., 2020; Fischer et al., 2019; Badreddine et al., 2022; Manhaeve et al., 2018). These methods are flexible and largely domain-agnostic, but they do not provide guarantees of exact rule satisfaction. To address hard constraints, a notable exception is MultiplexNet (Hoernle et al., 2022), which modifies the neural network architecture to embed disjunctive constraints and uses variational inference to train the model such that at least one disjunction is satisfied. However, MultiplexNet can only embed global constraints and cannot represent different constraints that depend on the input to the neural network. In addition, embedding general hard linear inequalities over both logical and continuous outputs represented as MILP has not been studied in the neuro-symbolic AI literature.

A major challenge in enforcing general MILP constraints is that the optimal solution of the associated optimization problem is typically non-differentiable with respect to the training parameters. As a result, gradients cannot be directly backpropagated to train the neural network. A simple workaround is to enforce feasibility only at inference time by solving an ILP constrained decoding problem on top of neural scores (Roth & Yih, 2005). More recent work aims to make discrete optimization modules differentiable. The Straight-through Estimator (STE) was initially proposed to differentiate the thresholding function by treating it as an identity function and was recently extended to combinatorial optimization problems with input-independent constraint set (Sahoo et al., 2023). One line of work considers convex relaxations of combinatorial optimization problems. For example, linear programming (LP) relaxations with regularization have been explored in (Wilder et al., 2019; Ferber et al., 2020; McKenzie et al., 2024), while semidefinite programming (SDP) relaxations are used in SATNet (Wang et al., 2019). However, these approaches cannot guarantee exact satisfaction of hard constraints. Another line of work derives smooth surrogate gradients for the argmax operator using stochastic perturbations (Berthet et al., 2020). While effective in certain settings, these methods require repeatedly solving combinatorial optimization problems to estimate one gradient, which can be computationally expensive. Moreover, they typically assume a fixed constraint set and do not support constraints that depend on the input.

A detailed comparison between existing approaches and our method is provided in Appendix A. Our work falls within the class of approaches that use convex relaxations to approximate mixed-integer linear constraints. The key distinction from prior work is that the relaxation we derive corresponds to the convex hull of the original constraint set, that is, the tightest possible convex relaxation. Owing to this tightness result, we can formally prove that the constraints are satisfied exactly. In contrast, existing methods such as (Wilder et al., 2019; Ferber et al., 2020; McKenzie et al., 2024) employ convex relaxations primarily as heuristics, and constraint satisfaction is not guaranteed in general. In summary, to the best of our knowledge, our method is the only approach that can enforce hard, input-dependent mixed-

integer linear constraints within neural networks during both training and inference.

## 3. Method

Let $x \in \mathcal{X}$ denote an input. A neural network $f_\theta : \mathcal{X} \to \mathcal{Y}$ produces an unconstrained prediction $\hat{y} = f_\theta(x)$, where $\mathcal{Y} \subseteq \mathbb{R}^d$ denotes the output space, which may include both continuous and discrete components. The goal of the proposed method is to enforce constraints on the prediction by projecting the unconstrained output $\hat{y}$ onto a feasible set defined by several constraints. Section 3.1 defines the class of constraints that the method can enforce and shows that they are as expressive as MILP and QF-LRA. Section 3.2 describes how the constraint set can be convexified to construct an LP-representable differentiable optimization layer.

### 3.1. Definition and expressiveness of the rules

**Input-dependent rules** We model domain knowledge using a collection of $R$ rules, each expressed as an implication

$$\mathbb{I}[x \in \mathcal{A}_r] \;\Rightarrow\; \mathbb{I}[y \in \mathcal{C}_r(x)], \qquad r = 1, \ldots, R, \quad (1)$$

where $\mathcal{A}_r \subseteq \mathcal{X}$ is the *activation set* of rule $r$, and $\mathcal{C}_r(x) \subseteq \mathcal{Y}$ is the corresponding *output feasibility set*. That is, whenever the input $x$ lies in $\mathcal{A}_r$, the output $y$ is required to belong to $\mathcal{C}_r(x)$. The total number of rules is denoted by $R$, and different subsets of rules may be active for different inputs.

Let

$$\mathcal{R}(x) := \{r \in \{1, \ldots, R\} \mid x \in \mathcal{A}_r\}$$

denote the set of rules activated by a given input $x$. The rule-induced feasible set for input $x$ is then given by the intersection of the corresponding output feasibility sets:

$$\mathcal{F}(x) \;=\; \bigcap_{r \in \mathcal{R}(x)} \mathcal{C}_r(x). \qquad (2)$$

This formulation captures domain knowledge expressed as multiple input-dependent if-then rules, where feasibility of the output depends explicitly on the input.

**Expressiveness of $\mathcal{A}_r$ and $\mathcal{C}_r(x)$** We assume that each activation set $\mathcal{A}_r$ is a compact subset of the input space $\mathcal{X}$, such as a polyhedral or ellipsoidal set, and that membership of $x$ in $\mathcal{A}_r$ can be efficiently verified.

We further assume that each output feasibility set $\mathcal{C}_r(x)$ can be represented as a finite union of polyhedral sets, given by

$$\mathcal{C}_r(x) = \bigcup_{j=1}^{m_r} \mathcal{S}_{rj}(x), \quad \mathcal{S}_{rj}(x) = \{y : A_{rj}(x)y \le b_{rj}(x)\}. \qquad (3)$$

where $m_r$ denotes the number of polyhedral sets associated with rule $r$, and $A_{rj}(x)$ and $b_{rj}(x)$ define the constraint

matrix and right-hand side vector, respectively. Both $A_{rj}(x)$ and $b_{rj}(x)$ may depend on the input $x$, allowing the feasible set to vary with the input. Each union of polyhedra, $\mathcal{C}_r(x)$, is called a *disjunction*. Each polyhedron within the union, $\mathcal{S}_{rj}(x)$, is called a *disjunct*.

Therefore, the set $\mathcal{F}(x)$ can represent the intersection of unions of polyhedra. The following proposition shows that $\mathcal{F}(x)$ can be as expressive as MILP.

**Proposition 3.1** (MILP and equivalent disjunctive constraint)**.** *Under mild boundedness assumptions, any MILP can be represented equivalently as the intersection of unions of polyhedra and vice versa.*

A proof can be found in Appendix B.1.

The next theorem shows that $\mathcal{F}(x)$ can also express any Quantifier-Free Linear Real Arithmetic (QF-LRA).

**Theorem 3.2** (Expressiveness of rule-induced feasible sets $\mathcal{F}(x)$ (informal))**.** *For a fixed input $x$, consider any constraint on the output $y \in \mathbb{R}^d$ that can be written using linear inequalities in $y$ and logical operators such as* and, or, *and* not, *where the coefficients of the inequalities may depend on $x$. Then the set of all outputs satisfying these constraints can be represented as a finite union of polyhedral sets in $y$.*

The formal version of the theorem and the corresponding proof are shown in Appendix B.2.

These expressiveness results show that the proposed framework unifies two complementary notations on constraint enforcement. Logical rules expressed using linear inequalities and Boolean operators, as studied in neural-symbolic AI, are covered by Theorem 3.2, while optimization-based constraint enforcement via MILPs, common in differentiable optimization layers, is captured by Proposition 3.1.

### 3.2. Differentiable convexified optimization layer

**Projection layer.** Given an input $x \in \mathcal{X}$ and the rule-feasible set $\mathcal{F}(x)$, we obtain a rule-consistent prediction from the unconstrained network output $\hat{y} = f_\theta(x)$, by computing the $\ell_1$-closest feasible point,

$$y^\star(x) \;\in\; \arg\min_{y \in \mathcal{F}(x)} \|y - \hat{y}\|_1. \qquad (4)$$

We use the $\ell_1$ distance because it admits an exact linear epigraph form that can be formulated as a linear program. This is crucial for our exactness results shown next in Theorem 3.6.

**$\ell_1$ epigraph reformulation.** We begin by rewriting the $\ell_1$ projection objective using a standard epigraph formulation. Introduce auxiliary variables $\eta \in \mathbb{R}_{\ge 0}^d$ and define

$$\mathcal{E}(\hat{y}) := \{(y, \eta) \in \mathbb{R}^d \times \mathbb{R}_{\ge 0}^d : -\eta \le y - \hat{y} \le \eta\}. \qquad (5)$$

Then the projection problem (4) is equivalent to the linear program

$$(y^\star(x), \eta^\star(x)) \in \arg\min_{(y,\eta)} \mathbf{1}^\top \eta$$
$$\text{s.t.} \quad (y, \eta) \in \mathcal{E}(\hat{y}), \quad y \in \mathcal{F}(x), \tag{6}$$

where $\mathcal{F}(x)$ denotes the (input-dependent) feasible set induced by the logical rules, as defined earlier in Eq. (2) .

**Global constraints.** In addition to logical rules, our framework supports global linear constraints on the output. Let $\mathcal{G}(x) \subseteq \mathbb{R}^d$ denote the global constraint set.

**Disjuncts with the epigraph formulation and global constraints** To derive our final exactness results, the global constraint set $\mathcal{G}(x)$ and the epigraph constraints $\mathcal{E}(\hat{y})$ must be incorporated *inside each disjunct*. Accordingly, for each polyhedral set $\mathcal{S}_{rj}(x)$, we extend it to incorporate the $\eta$ variables and associated constraints. The resulting set is defined as

$$\widehat{\mathcal{S}}_{rj}(x; \hat{y}) := \Big\{(y, \eta) : A_{rj}(x)y \leq b_{rj}(x),$$
$$(y, \eta) \in \mathcal{E}(\hat{y}), \ y \in \mathcal{G}(x)\Big\}. \tag{7}$$

The set $\mathcal{C}_r(x)$ and $\mathcal{F}(x)$ can be extended to consider the epigraph variables and the global constraints as,

$$\widehat{\mathcal{C}}_r(x; \hat{y}) = \bigcup_{j=1}^m \widehat{\mathcal{S}}_{rj}(x; \hat{y}),$$
$$\widehat{\mathcal{F}}(x; \hat{y}) = \bigcap_{r \in \mathcal{R}(x)} \widehat{\mathcal{C}}_r(x; \hat{y}) \tag{8}$$

The projection layer can be written compactly as

$$(y^\star(x, \hat{y}), \eta^\star(x, \hat{y})) \in \arg\min_{(y,\eta)} \mathbf{1}^\top \eta$$
$$\text{s.t.} \quad (y, \eta) \in \widehat{\mathcal{F}}(x; \hat{y}), \tag{9}$$

The resulting set $\widehat{\mathcal{F}}(x; \hat{y})$ is however still non-convex due to the union across multiple disjuncts in each rule. Therefore, to allow differentiability, we apply a convex-hull relaxation $\text{conv}(\widehat{\mathcal{F}}(x; \hat{y}))$ which is the smallest convex set encompassing the set $\widehat{\mathcal{F}}(x; \hat{y})$. Before we formulate the convex hull relaxation for our problem, we provide the extended formulation that can represent the convex hull of the union of bounded polyhedra. The idea is to first provide the mathematical tool for convexifying each individual set $\widehat{\mathcal{C}}_r(x; \hat{y})$ before convexifying their intersection.

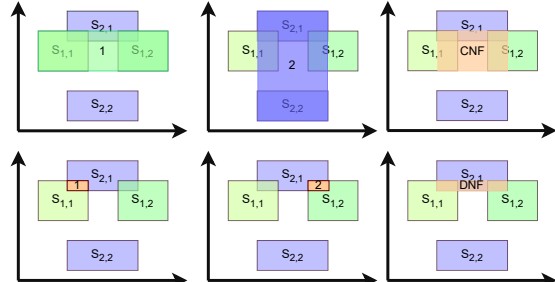

*Figure 1.* CNF and DNF illustration for 2 rules with 2 disjuncts each. The nonconvex set we are trying to convexify is $(\mathcal{S}_{1,1} \cup \mathcal{S}_{1,2}) \cap (\mathcal{S}_{2,1} \cup \mathcal{S}_{2,2})$ **CNF** (top): Takes the convex hull of $(\mathcal{S}_{1,1} \cup \mathcal{S}_{1,2})$ (top left) and the convex hull of $(\mathcal{S}_{2,1} \cup \mathcal{S}_{2,2})$ (top middle) and intersect them (top right). **DNF** (bottom): Intersects the combinations of the disjuncts, $\mathcal{S}_{1,1} \cap \mathcal{S}_{2,1}$ (bottom left), $\mathcal{S}_{1,1} \cap \mathcal{S}_{2,2}$ (empty), $\mathcal{S}_{1,2} \cap \mathcal{S}_{2,1}$ (bottom middle), $\mathcal{S}_{1,2} \cap \mathcal{S}_{2,2}$ (empty). The DNF takes the convex hull of all the four intersections (bottom right). The DNF is a true subset of the CNF and represents the convex hull of the nonconvex set.

**Convex-hull reformulation of disjunctions** The following proposition constructs the convex hull of the union of bounded polyhedra in a lifted space, by making *copies* of the variables representing each of the possible disjunct.

**Proposition 3.3** (Extended formulation for $\text{conv}(\cup_j \mathcal{S}_j)$). *Let bounded polyhedral sets $\mathcal{S}_j = \{z : A_j z \leq b_j\}$ for $j = 1, \ldots, m$, where $z$ denotes the lifted variable $(y, \eta)$. Then*

$$\text{conv}\Big(\bigcup_{j=1}^m \mathcal{S}_j\Big) = \Big\{w \mid \exists \{w_j\}_{j=1}^m, \ \{\lambda_j\}_{j=1}^m :$$
$$A_j w_j \leq \lambda_j b_j \quad \forall j, \quad w = \sum_{j=1}^m w_j,$$
$$\sum_{j=1}^m \lambda_j = 1, \quad \lambda_j \geq 0 \quad \forall j\Big\}. \tag{10}$$

*where $w_j$ are* copies *of the lifted variable associated with each disjunct, and $\lambda$ represent the weight of the convex combination. See Theorem 2.1 (Balas, 2018) for exact proof.*

Setting $\lambda_i = 1$ recovers the $i$-th disjunct exactly. In contrast to binary selection variables used in MILP formulations, the continuous variables $\lambda$ allow convex combinations of disjuncts, yielding a linear relaxation. This extended formulation in Proposition 3.3 gives us the convex hull for each set $\widehat{\mathcal{C}}_r(x; \hat{y})$ individually.

**Multiple active rules.** We further consider how to convexify the set $\widehat{\mathcal{F}}(x; \hat{y})$, which is the intersection of multiple unions of polyhedral set $\widehat{\mathcal{C}}_r(x; \hat{y})$. The *conjunctive normal*

*form (CNF)* relaxation convexifies each rule independently and then intersects the resulting convex sets:

$$\widetilde{\mathcal{F}}_{\mathrm{CNF}}(x;\hat{y}) = \bigcap_{r \in \mathcal{R}(x)} \mathrm{conv}\big(\widehat{\mathcal{C}}_r(x;\hat{y})\big). \qquad (11)$$

Note that in general, $\widetilde{\mathcal{F}}_{\mathrm{CNF}}(x;\hat{y})$ is not the convex hull of $\hat{\mathcal{F}}(x;\hat{y})$ because the convex hull operator and the intersection are not interchangeable. If each $\widehat{\mathcal{C}}_r(x;\hat{y})$ is nonconvex, we have $\mathrm{conv}\big(\hat{\mathcal{F}}(x;\hat{y})\big) \subset \widetilde{\mathcal{F}}_{\mathrm{CNF}}(x;\hat{y})$.

To obtain the convex hull of all the active rules, $\mathrm{conv}\big(\hat{\mathcal{F}}(x;\hat{y})\big)$, the *disjunctive normal form (DNF)* relaxation distributes intersections over unions by enumerating all possible *intersections* formed by selecting one polyhedral set from each active rule. Let $\Pi(x) := \prod_{r \in \mathcal{R}(x)} [m_r]$ denote the set of such selections (each $k \in \Pi(x)$ is a tuple $k = (k_r)_{r \in \mathcal{R}(x)}$ with $k_r \in \{1, \ldots, m_r\}$). Then

$$\widetilde{\mathcal{F}}_{\mathrm{DNF}}(x;\hat{y}) = \mathrm{conv}\left( \bigcup_{k \in \Pi(x)} \bigcap_{r \in \mathcal{R}(x)} \widehat{\mathcal{S}}_{r,k_r}(x;\hat{y}) \right). \quad (12)$$

Note that each of the intersection $\bigcap_{r \in \mathcal{R}(x)} \widehat{\mathcal{S}}_{r,k_r}(x;\hat{y})$ is a polyhedron since it is an intersection of a finite number of polyhedra. The intersection can be easily represented by combining all the linear inequalities in each of the sets $\widehat{\mathcal{S}}_{r,k_r}(x;\hat{y})$, $\forall r \in \mathcal{R}(x)$. Eq (12) can be seen as the convex hull of a finite union of polyhedra. Therefore, we can apply the extended formulation for a finite union of polyhedra defined in Proposition 3.3 to obtain an LP representation of $\widetilde{\mathcal{F}}_{\mathrm{DNF}}(x;\hat{y})$. It can be shown $\widetilde{\mathcal{F}}_{\mathrm{DNF}}(x;\hat{y}) = \mathrm{conv}\big(\hat{\mathcal{F}}(x;\hat{y})\big)$, i.e., the DNF relaxation gives the exact convex hull of the projection problem in Eq (9). This exactness is formalized in Theorem 3.6 and proved in Appendix B.3.

*Remark* 3.4 (Scalability-tightness trade-off). The CNF relaxation introduces one convex-hull formulation per active rule and therefore scales linearly with $|\mathcal{R}(x)|$. In contrast, the DNF relaxation requires enumerating all combinations of polyhedral regions across active rules, resulting in a number of terms that grows exponentially in $|\mathcal{R}(x)|$. As a result, CNF provides a scalable but weaker relaxation, while DNF yields a tighter relaxation of the feasible set at the expense of higher computational complexity. An illustrative example comparing the DNF and CNF is shown in Figure 1.

**Corollary 3.5.** *If each $\widehat{\mathcal{C}}_r(x;\hat{y})$ is convex for all active rules $r \in \mathcal{R}(x)$, then*

$$\widetilde{\mathcal{F}}_{\mathrm{CNF}}(x;\hat{y}) = \widetilde{\mathcal{F}}_{\mathrm{DNF}}(x;\hat{y}).$$

**Theorem 3.6** (Exact rule satisfaction under DNF projection). *When solving the problem in Eq. (9) with $\widetilde{\mathcal{F}}_{\mathrm{DNF}}(x;\hat{y})$ using the extended LP formulation, the extreme point solution to the LP satisfies the constraints of the original formulation Eq (9).*

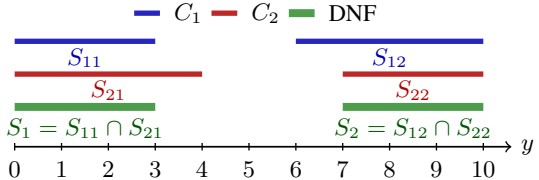

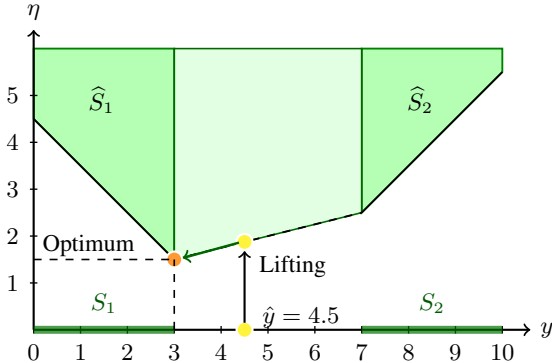

*Figure 2.* **Lifted projection for two active rules.** The top number line shows two active rules, $C_1 = S_{11} \cup S_{12}$ and $C_2 = S_{21} \cup S_{22}$, whose DNF intersections give $S_1 \cup S_2 = [0,3] \cup [7,10]$. Following Eqs. (5)-(12), these DNF terms are lifted into $(y, \eta)$-space as $\widehat{S}_1$ and $\widehat{S}_2$. For $\hat{y} = 4.5$, minimizing $\eta$ over the lifted convex hull returns the extreme point $y^\star = 3$, which satisfies both rules exactly.

A formal version of the theorem and the proof can be found in Appendix B.3.

*Remark* 3.7. In particular, if the lifted LP is solved with a solver returning an extreme point (e.g., using the simplex algorithm), the returned solution satisfies all original rules.

*Remark* 3.8. An analogous guarantee does not generally hold for the CNF relaxation: extreme points of $\widetilde{\mathcal{F}}_{\mathrm{CNF}}(x;\hat{y})$ need not correspond to extreme points of the convex hull of the original set.

Figure 2 illustrates the exact-satisfaction mechanism for two active disjunctive rules. The top number line shows how the DNF expansion keeps only the consistent intersections of the active disjuncts, yielding the nonconvex feasible set $S_1 \cup S_2 = [0,3] \cup [7,10]$. The key step is that the epigraph constraints defining the $\ell_1$ projection distance are introduced before convexification, as in Eqs. (5)-(12). Thus, the DNF terms are lifted into $(y, \eta)$-space, and the projection minimizes the vertical coordinate $\eta$ over the lifted convex hull rather than over a naive convexification in the original $y$-space. Geometrically, this sends the infeasible prediction toward the lowest point of the lifted hull. When the LP optimum is an extreme point, it corresponds to one original DNF term and therefore gives a prediction satisfying all active rules. Appendix D provides the full derivation.

**Sequential convexification and basic step.** The CNF and DNF relaxations represent two extremes in the trade-off

between model size and tightness. In many applications, only a subset of rules exhibits strong interactions that benefit from a DNF expansion. This motivates a *sequential* convexification scheme (Balas, 2018), that expands disjunctions for a selected subset of rules.

Let $\mathcal{R}(x) = \mathcal{R}_D(x) \cup \mathcal{R}_C(x)$ be a partition of the active rules, where $\mathcal{R}_D(x)$ are expanded in DNF form and $\mathcal{R}_C(x)$ are kept in CNF form. The resulting relaxation denoted by $\widetilde{\mathcal{F}}_{\text{pDNF}}(x; \hat{y}, \mathcal{R}_C(x), \mathcal{R}_D(x))$ is

$$\widetilde{\mathcal{F}}_{\text{pDNF}}(x; \hat{y}, \mathcal{R}_C(x), \mathcal{R}_D(x)) =$$
$$\bigcap_{r \in \mathcal{R}_C(x)} \text{conv}(\widehat{\mathcal{C}}_r(x; \hat{y})) \cap \tag{13}$$
$$\text{conv}\left( \bigcup_{k \in \Pi(x)} \bigcap_{r \in \mathcal{R}_D(x)} \widehat{\mathcal{S}}_{r,k_r}(x; \hat{y}) \right).$$

This sits in between CNF and DNF in terms of model complexity and relaxation tightness. A *basic step* is defined as converting a given relaxation $\widetilde{\mathcal{F}}_{\text{pDNF}}(x; \hat{y}, \mathcal{R}_C(x), \mathcal{R}_D(x))$ to $\widetilde{\mathcal{F}}_{\text{pDNF}}(x; \hat{y}, \mathcal{R}_C(x) \backslash \{r'\}, \mathcal{R}_D(x) \cup \{r'\})$, i.e., it adds an additional rule to the DNF form. In practice, basic steps can be sequentially applied to convexify the CNF.

*Remark* 3.9. Currently, there is no well-defined notion of an "optimal" sequence in which to apply the basic steps. In the experiments section, we illustrate the tradeoff between tightness and tractability by applying a fixed sequence of basic steps determined by a predefined lexicographic order of the rules. A more systematic investigation of how to choose which rules to intersect at each iteration is left for future work.

**Corollary 3.10** (Ordering and special cases)**.** *For any input $x$ and any such partition,*

$$\widetilde{\mathcal{F}}_{\text{DNF}}(x; \hat{y}) \subseteq \widetilde{\mathcal{F}}_{\text{pDNF}}(x; \hat{y}) \subseteq \widetilde{\mathcal{F}}_{\text{CNF}}(x; \hat{y}).$$

*Moreover, $\mathcal{R}_D(x) = \emptyset$ recovers CNF, $\mathcal{R}_D(x) = \mathcal{R}(x)$ recovers DNF.*

**Projection as a differentiable linear program** For each fixed input $x$, the projection to the convex hull relaxation of the CNF/DNF/pDNF can be written as an LP in extended form, with $\hat{y}$ entering only through the $\ell_1$ epigraph constraints (see formulation in proof of Theorem 3.6 in Appendix B.3). As an illustration, in the DNF convex-hull formulation, we introduce disjunct-specific copies $(y_k, \eta_k)$ and the convex combination weights $\lambda_k$, and the epigraph constraints take the form

$$\eta_k \geq y_k - \lambda_k \hat{y}, \qquad \eta_k \geq \lambda_k \hat{y} - y_k, \tag{14}$$

together with the aggregation

$$\sum_k y_k = y, \quad \sum_k \lambda_k = 1 \tag{15}$$

Note that Eqs (14) and (15) are the only two places the unconstrained prediction from the NN, $\hat{y}$, appears. To train the deep learning model end-to-end, we need to differentiate the optimal solution to the linear program, $y^\star(x; \hat{y})$, with respect to parameter $\hat{y}$. The gradient is computed by implicit differentiation of the KKT conditions in open-source software like CVXPYlayer (Agrawal et al., 2019) and DiffOpt.jl (Besançon et al., 2023)). A concise summary of the complete forward and backward pass is provided in Appendix C. The code is open sourced as a general use package `DisjunctiveNet.jl` [1].

**Improving differentiability of the optimization layer** One caveat here is that the parameters $\hat{y}$ appear in the left hand side of the LP, which may cause the optimal solution $y^\star(x; \hat{y})$ to change discontinuously. To resolve this issue, a common approach is to add a quadratic regularizer and make the problem strongly convex. This approximation has been used in multiple previous works to make LP differentiable (Sadana et al., 2025). We use the extended form LP as the forward pass and the gradient from the regularized problem (a strongly convex quadratic program) in the backward pass.

## 4. Experiments

We evaluate our method against several baselines under *input-dependent logical constraints* on (i) a synthetic cooling-control task and (ii) single-cell RNA sequencing (scRNA-seq) classification with marker-gene rules.

**Baselines.** A base NN acts as an unconstrained model unaware of any rules. A penalty-based NN (pen) adds penalty terms for all active rules to the loss function, which acts as a soft method for rule enforcement. We also include a finetuned penalty baseline (fine-pen), which is initialized from the trained base NN and then finetuned with the same penalty terms, providing a direct comparison to projection-based finetuning under the same pretrained initialization. All projection-based models (CNF and DNF), as described Section 3.2, are initialized from the trained base NN. The models are then finetuned with the projection layer, allowing learning to focus on integrating rule enforcement rather than re-learning the underlying prediction task.

All experiments were implemented in `Julia`. Neural network architectures and training algorithms were implemented using `Flux.jl` (Innes et al., 2018). Differentiable projection layers were realized via LP-based projection operators implemented through the QP interface of `DiffOpt.jl` (Besançon et al., 2023), which performs implicit differentiation using KKT conditions. All methods were evaluated under a fixed configuration without per-

---

[1] https://github.com/li-group/DisjunctiveNet.jl.git

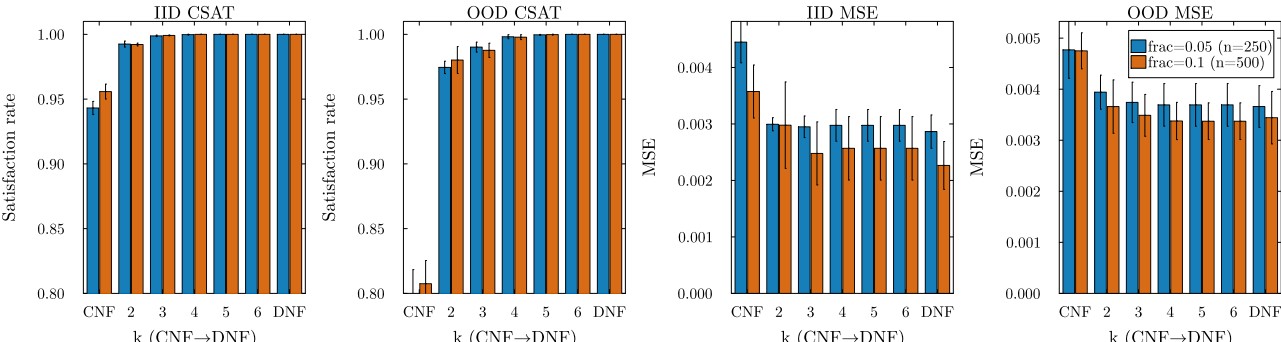

*Figure 3.* **Synthetic task: Sequential convexification.** Tightening the relaxation from CNF toward DNF improves constraint satisfaction (CSAT) and improves MSE, across two training set sizes. The CNF achieves significantly less constraint satisfaction in the OOD test set, and higher MSE compared to DNF and the intermediates. The plateau in CSAT is observed due to the fact that very few samples have such a large number of active rules.

method hyperparameter tuning.

**Metrics.** We report predictive performance (MSE for the synthetic task; macro-F1 for scRNA-seq) and *constraint satisfaction* (CSAT), defined as the fraction of samples whose *active* input-dependent rules are satisfied by the final model outputs. For the scRNA CSAT, we only consider samples that have no contradicting active rules. Results are reported as mean $\pm$ std across 3 random runs.

### 4.1. Synthetic cooling-control problem

**Task and data.** The task models a synthetic cooling control dataset. The model predicts continuous control actions (fan speed, chiller usage, pump power) based on environmental and operational inputs, subject to global operating constraints and multiple simultaneously active disjunctive safety/operational rules. Ground-truth targets are produced by an oracle Quadratic Program (QP) that also enumerates feasible rule-disjunct assignments for the active rules and selects a minimum-cost solution. The NN aims to approximate the QP objective. We then add small perturbations to the ground truth obtained to mimic measurement variability and randomness. The train and *IID* test set are derived from the same distribution while the *OOD* test set is generated from a shifted distribution. Additional details about the rules and test set distributions are mentioned in Appendix E.2

**Sequential convexification.** We perform this experiment, to study the constraint tightness as we sequentially convexify (Section. 3.2) the feasible set by performing repeated basic steps. We start with the CNF and increase the number of rules considered in the DNF expansion. We obtain the DNF relaxation when convexifying all the rules (7 here). We perform this experiment for two training set sizes ($n = 250, 500$) and report the CSAT and MSE for both *IID* and *OOD* test sets.

As shown in Fig. 3, with sequential convexification, CSAT improves monotonically and rapidly approaches DNF-level feasibility in both *IID* and *OOD* settings, demonstrating that expanding even a small number of disjunctions captures most of the logical structure required for rule satisfaction. Additionally, most samples do not necessarily have all possible rules as active. MSE exhibits improvement in predictive performance with convexification, and similar to CSAT, plateaus as we move towards DNF. These results highlight a practical accuracy-scalability trade-off: a suitable sequential scheme can recover most of the feasibility benefits of DNF at substantially lower model complexity and computational overhead. It is important to note here that choosing which rules to convexify is an open challenge. Here, we simply perform convexification on a fixed rank order.

**Dataset-size scaling.** Finally, we evaluate how our method compares to the baselines as the amount of training data varies. We aim to answer when satisfying input-dependent logical constraints provides the greatest benefit in predictive performance. All methods are trained with increasing numbers of training samples and evaluated on fixed *IID* and *OOD* test sets.

Figure 4 shows MSE and CSAT as functions of training set size. Since lower MSE is better, both CNF and DNF projection layers substantially outperform the unconstrained base model, the standard penalty baseline, and the finetuned penalty baseline across training set sizes. The improvement is especially pronounced under the OOD shift, where incorporating the rules through projection yields much lower MSE than either unconstrained learning or soft penalty-based enforcement. This suggests that the projection layer provides a strong inductive bias that improves generalization, rather than merely enforcing feasibility at inference time.

In terms of CSAT, DNF achieves complete constraint satisfaction across both IID and OOD test sets, while CNF main-

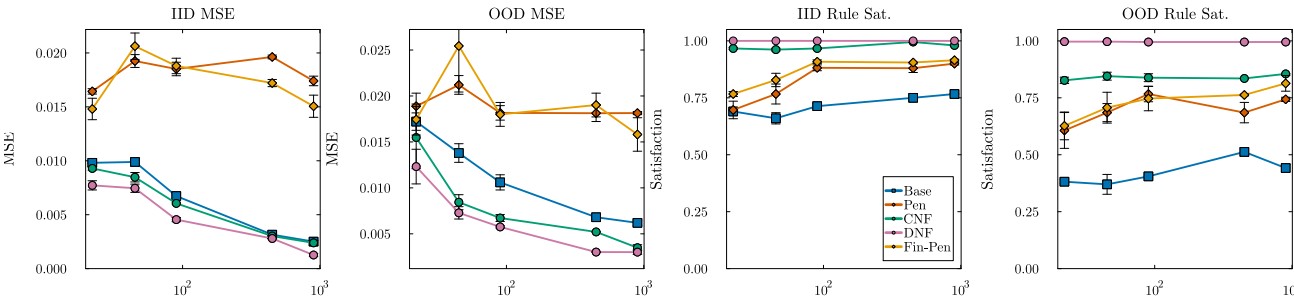

*Figure 4.* **Synthetic task: Dataset size.** Performance of different methods for increasing amounts of training data. Projection-based methods (CNF, DNF) achieve substantially lower MSE and higher rule satisfaction than the base and penalty-based baselines. DNF achieves complete constraint satisfaction across both IID and OOD test sets.

*Table 1.* Computational overhead on the synthetic cooling task. Inference time is measured per sample. LP sizes are reported over samples with at least two active rules.

| Model | Inf time | Variables | Constraints |
|---|---|---|---|
| Base | $< 1\,\mu\mathrm{s}$ | – | – |
| Penalty | $< 1\,\mu\mathrm{s}$ | – | – |
| CNF | $25.03\,\mathrm{ms}$ | $37.50 \pm 6.54$ | $137.28 \pm 26.29$ |
| DNF | $28.62\,\mathrm{ms}$ | $44.39 \pm 14.86$ | $160.53 \pm 62.69$ |

tains consistently high satisfaction. The base and penalty-based methods improve modestly with additional data, but remain substantially below the projection-based methods, particularly in the OOD setting. The finetuned penalty baseline benefits from the same pretrained initialization as the projection models, but still does not reliably enforce the constraints. These results show that differentiable projection layers are particularly effective when rules encode structural knowledge that is difficult to learn from data alone: they achieve both stronger predictive performance and substantially higher rule satisfaction than unconstrained or penalty-based training.

**Computational overhead.** Table 1 summarizes the projection-layer overhead on the synthetic cooling task. The unconstrained and penalty models have negligible inference overhead, while the CNF and DNF projection layers require solving an LP at inference time. Even with multiple active rules, both formulations remain practical in this setting: CNF and DNF take 25.03 ms and 28.62 ms per sample, respectively. DNF produces larger LPs because it enumerates intersections of active disjuncts, but the realized size is often much smaller than the worst-case exponential bound because many disjunct combinations are infeasible or inactive. A more detailed breakdown of inference time and LP sizes, including statistics for samples with one or more active rules, is provided in Appendix E.3.

### 4.2. Case study: scRNA-seq classification

**Task and rules.** Next, we consider the application of the proposed method to a classification task. Single-cell RNA sequencing (scRNA-seq) is a biomedical task that uses gene expression counts $x$ to classify cell types $y$. Such data are expensive to generate and inherently high-dimensional and sparse. We perform scRNA cell-type classification using the PBMC3k dataset, a standard benchmark consisting of peripheral blood mononuclear cells annotated into discrete cell types. The task is to predict class probabilities for each cell $y \in \Delta^8$, given normalized gene expression features. Domain knowledge is encoded through input-dependent marker-gene rules of the form

$$G_r(u) \geq \tau_r \;\Rightarrow\; \bigvee_{c \in \mathcal{C}_r} y_c \geq \rho_r, \qquad (16)$$

where $G_r(u)$ is a linear function of gene expression, $\mathcal{C}_r$ denotes a set of candidate cell types associated with the marker, and $\tau_r, \rho_r$ are fixed thresholds. When the antecedent is satisfied, the rule enforces that at least one associated cell type must receive sufficiently high predicted probability. In addition to the baselines defined before, we also run a rules baseline, that randomly selects a feasible disjunction from the active rules. This acts as an indication of the quality and accuracy of the rules.

Marker-gene rules may be noisy or partially contradictory: for a given sample, multiple active rules can induce constraints whose feasible sets have an empty intersection. In such cases, no output can simultaneously satisfy all active rules; when this occurs, we bypass the projection and pass the NN output and gradients through unchanged to preserve well-defined training dynamics. For the CSAT reporting, we exclude those samples as our aim is to understand how often the different methods can satisfy the original constraints whenever possible.

**Results.** Figure 5 reports macro-F1 (top) and CSAT (bottom) as functions of training set size. The exact quantitative

results are provided in Appendix E.5. Projection layers substantially improve rule satisfaction compared to the base network and penalty-based baselines across all training set sizes. DNF-based projections achieve the highest rule satisfaction, while CNF also significantly improves satisfaction over unconstrained and penalty-based training, though at a consistently lower rate than DNF.

In terms of accuracy, the projection layers are most beneficial in the low-data regime, where incorporating biologically motivated marker-gene rules provides a useful inductive bias and improves macro-F1 relative to the base model. As the training set grows, the base model becomes increasingly competitive and achieves the highest F1, indicating that sufficient labeled data can partially compensate for missing structural knowledge in terms of predictive accuracy. However, this improved accuracy comes with substantially lower rule satisfaction compared to the projection-based models. The finetuned penalty model improves over the standard penalty baseline in rule satisfaction, but remains below CNF and DNF, illustrating that soft penalties do not reliably enforce the rules even when initialized from the same pretrained base model. The rules-only baseline performs substantially worse in F1 despite high rule satisfaction, indicating that rules alone do not suffice for accurate classification and must be combined with data-driven learning. Overall, the projection methods are most useful when data are scarce or when satisfying the constraints is non-negotiable, while revealing a clear trade-off between predictive accuracy and exact rule adherence in higher-data regimes.

## 5. Conclusions

We presented a projection-based framework for enforcing input-dependent mixed logical-numeric constraints in neural networks, providing exact feasibility guarantees during both training and inference whenever the constraint system is feasible. The framework supports general MILP-representable constraints and logical implications by reformulating them as disjunctive structures, and integrates naturally into end-to-end learning without relying on surrogate penalties or post-hoc correction. The framework is model-agnostic and can seamlessly fit into other neural architectures. Empirical results on a synthetic cooling problem and scRNA-seq classification with marker-gene rules demonstrate that projection layers substantially improve rule adherence and predictive performance in low-data regimes, while revealing a principled trade-off between constraint tightness and accuracy.

Several directions remain for future work. A key extension is to incorporate convex nonlinear constraints within disjuncts, enabling richer domain knowledge while preserving the logical structure of the rules. This would also allow extending the $\ell_1$-norm projection problem to the Euclidean norm, leveraging results from the nonlinear disjunctive pro-

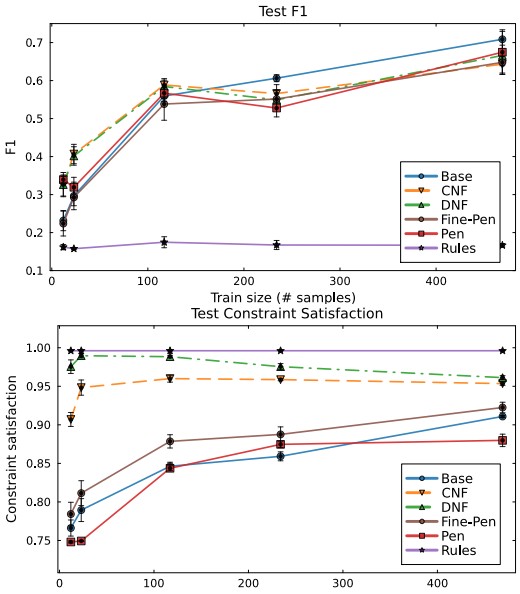

*Figure 5.* **PBMC3k scRNA-seq results vs. training set size (test set). Top:** F1. Enforcing rules via projection improves performance in low-data regimes; tighter DNF-based projections can trade a small amount of F1 for stronger rule adherence. **Bottom:** sample-level rule satisfaction. DNF reaches 100%, indicating that essentially all samples with *non-contradictory* active rules are satisfied.

gramming literature (Ceria & Soares, 1999). In addition, the trade-off between computational complexity and constraint satisfaction for CNF and DNF formulations warrants further study. To this end, improved sequential convexification heuristics, such as those proposed in (Li & Grossmann, 2019), may enable adaptive transition between CNF and partial DNF formulations during neural network training.

**Limitations** A primary limitation of our framework is computational. While the CNF relaxation scales linearly with the number of active rules, the DNF relaxation can grow exponentially because it enumerates combinations of disjuncts across rules. In practice, this motivates the use of CNF or partial-DNF formulations when the active rule set becomes large. Empirical stress-test results for larger active rule sets, together with comparisons to rule counts in representative neuro-symbolic benchmarks, are reported in Appendix E.4. A second limitation is that the exact-satisfaction guarantee for DNF depends on the solver returning optimal extreme-point solutions of the lifted LP, as is the case for simplex-based solvers and barrier methods with crossover. Finally, in applications with noisy, misspecified, or contradictory rules, additional conflict-handling mechanisms may be required. We therefore view this framework as most appropriate for settings where the rule base is well validated and potential conflicts in domain knowledge can be carefully monitored.

## Impact Statement

This paper presents work whose goal is to advance the field of machine learning. There are many potential societal consequences of our work, none of which we feel must be specifically highlighted here.

## Acknowledgements

C.L. would like to acknowledge the financial support from NSF awards CBET- 2441184, DMS-2424004, ONR award N000142412641. S.P. was supported by the NSF award DMS-2424004.

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

# A. Detailed comparison with related works

We provide a clear comparison of the representative cited works in the related works section in Table 2.

- **Logic** Denotes whether a method can represent logical decisions. This includes Boolean variables studied in the neuro-symbolic AI literature as well as binary variables in approaches that embed mixed-integer programming problems as layers.

- **Linear** Denotes whether a method can embed general linear constraints of the form $Ax \leq b$. Methods that rely on activation functions to enforce only specialized classes of linear inequalities are not checked.

- **Mix** Denotes whether a method can represent constraints involving both logical and continuous variables, such as constraints that are representable by mixed-integer linear programs.

- **Hard** Denotes whether the method provides theoretical guarantees of exact constraint satisfaction during both training and inference.

- **Input-dep** Denotes whether a method can enforce constraints that depend on the input to the neural network.

| Method | Logic | Linear | Mix | Hard | Input dep |
|---|:---:|:---:|:---:|:---:|:---:|
| OptNet (Amos & Kolter, 2017) | ✗ | ✓ | ✗ | ✓ | ✓ |
| CVXPYlayers (Agrawal et al., 2019) | ✗ | ✓ | ✗ | ✓ | ✓ |
| DC3 (Donti et al., 2021) | ✗ | ✓ | ✗ | ✗ | ✓ |
| Primal dual (Park & Van Hentenryck, 2023) | ✗ | ✓ | ✗ | ✗ | ✓ |
| KKTHardNet (Iftakher et al., 2025) | ✗ | ✓ | ✗ | ✓ | ✓ |
| DecisionRuleNet (Constante-Flores et al., 2025) | ✗ | ✓ | ✗ | ✓ | ✓ |
| Homeomorphic projection (Liang et al., 2024) | ✗ | ✓ | ✗ | ✓ | ✓ |
| FSNet (Nguyen & Donti, 2025) | ✗ | ✓ | ✗ | ✓ | ✓ |
| Semantic Loss (Xu et al., 2018) | ✓ | ✓ | ✗ | ✗ | ✗ |
| DL2 (Fischer et al., 2019) | ✓ | ✓ | ✓ | ✗ | ✓ |
| LTN (Badreddine et al., 2022) | ✓ | ✗ | ✓ | ✗ | ✓ |
| DeepProbLog (Manhaeve et al., 2018) | ✓ | ✓ | ✗ | ✗ | ✓ |
| MultiplexNet (Hoernle et al., 2022) | ✓ | ✗ | ✓ | ✓ | ✗ |
| Straight through estimator (Sahoo et al., 2023) | ✓ | ✓ | ✓ | ✓ | ✗ |
| SATNet (Wang et al., 2019) | ✓ | ✗ | ✗ | ✗ | ✗ |
| LP relaxation of ILP (Wilder et al., 2019) | ✓ | ✓ | ✓ | ✗ | ✗ |
| MIPaaL (Ferber et al., 2020) | ✓ | ✓ | ✓ | ✗ | ✗ |
| DYS-Net (McKenzie et al., 2024) | ✓ | ✓ | ✓ | ✗ | ✗ |
| Perturbed optimizer (Berthet et al., 2020) | ✓ | ✓ | ✓ | ✓ | ✗ |
| CombOptNet (Paulus et al., 2021) | ✓ | ✓ | ✓ | ✗ | ✗ |
| ILP inference (Roth & Yih, 2005) | ✓ | ✓ | ✓ | ✗ | ✓ |
| DRL (Stoian & Giunchiglia, 2025) | ✓ | ✗ | ✓ | ✓ | ✗ |
| Ours | ✓ | ✓ | ✓ | ✓ | ✓ |

*Table 2.* Comparison of constraint-aware learning methods.

# B. Proof of propositions and theorems

### B.1. Proof of Proposition 3.1

*Proof.* Fix $x$ and abbreviate $A_{rj} = A_{rj}(x)$ and $b_{rj} = b_{rj}(x)$. Consider a single active rule $r$ with $\mathcal{C}_r(x) = \bigcup_{j=1}^{m_r} \mathcal{S}_{rj}(x)$ and $\mathcal{S}_{rj}(x) = \{y \in \mathbb{R}^d : A_{rj}y \leq b_{rj}\}$. Assume $y$ is constrained to a known bounded box $\mathcal{B} = \{y : \ell \leq y \leq u\}$.

Introduce binary variables $\delta_{rj} \in \{0, 1\}$ for $j = 1, \ldots, m_r$ with $\sum_{j=1}^{m_r} \delta_{rj} = 1$. For each inequality row $i$ in $A_{rj}y \leq b_{rj}$, choose a constant $M_{rji}$ valid over $y \in \mathcal{B}$ such that

$$(A_{rj}y)_i \leq (b_{rj})_i + M_{rji} \qquad \text{for all } y \in \mathcal{B}.$$

Then the disjunctive constraint $y \in \mathcal{C}_r(x)$ is equivalent (over $y \in \mathcal{B}$) to the mixed-integer linear system

$$\sum_{j=1}^{m_r} \delta_{rj} = 1, \qquad \delta_{rj} \in \{0,1\} \ \forall j,$$

$$(A_{rj}y)_i \le (b_{rj})_i + M_{rji}(1 - \delta_{rj}), \qquad \forall j \in \{1,\dots,m_r\}, \ \forall i.$$

If $\delta_{rj} = 1$ for some $j$, the corresponding constraints reduce to $A_{rj}y \le b_{rj}$, i.e., $y \in \mathcal{S}_{rj}(x)$; if $\delta_{rj} = 0$, the constraints are relaxed and become redundant on $\mathcal{B}$ by construction of $M_{rji}$. Conversely, if $y \in \mathcal{S}_{rj}(x)$ for some $j$, setting $\delta_{rj} = 1$ and $\delta_{rj'} = 0$ for $j' \ne j$ yields feasibility. This is the standard disjunctive-programming (big-$M$) formulation for finite unions of polyhedra on bounded domains; see, e.g., Balas (2018); Jeroslow (1987).

To represent $\mathcal{F}(x) = \bigcap_{r \in \mathcal{R}(x)} \mathcal{C}_r(x)$, we introduce an independent selector vector $\delta_r := (\delta_{r1},\dots,\delta_{rm_r})$ for each active rule $r \in \mathcal{R}(x)$ and impose the above system simultaneously for all such $r$, sharing the same decision variable $y$. The resulting constraint system is a single MILP in variables $(y, \{\delta_r\}_{r \in \mathcal{R}(x)})$ whose projection onto the $y$-coordinates is exactly $\mathcal{F}(x)$. The total number of binary variables is $\sum_{r \in \mathcal{R}(x)} m_r$, one per disjunct in each active rule. $\qquad\square$

## B.2. Proof of Theorem 3.2

**Theorem B.1** (Quantifier-free linear-atomic formulas yield finite unions of polyhedra). *Let $\varphi(x,y)$ be a quantifier-free formula with free variable $y \in \mathbb{R}^d$, built from linear inequalities in $y$ whose coefficients may depend on $x$, together with Boolean connectives $(\wedge, \vee, \neg)$. Assume predicates are drawn from the class*

$$a(x)^\top y \le b(x) \quad and \quad a(x)^\top y \ge b(x),$$

*for given functions $a : \mathcal{X} \to \mathbb{R}^d$ and $b : \mathcal{X} \to \mathbb{R}$. Let $\mathcal{Y} \subseteq \mathbb{R}^d$ denote the (bounded) output domain.*

*Then for every $x \in \mathcal{X}$ there exists an integer $m \ge 1$ and a consequent set*

$$\mathcal{C}(x) = \bigcup_{j=1}^{m} \mathcal{S}_j(x), \qquad \mathcal{S}_j(x) = \{\, y \in \mathcal{Y} : A_j(x)y \le b_j(x)\,\},$$

*such that*

$$\{\, y \in \mathcal{Y} : \varphi(x,y)\,\} = \mathcal{C}(x).$$

*Proof.* Let $x \in \mathcal{X}$ be arbitrary and treat all coefficients evaluated at $x$ as constants. Convert $\varphi(x,y)$ to *negation normal form*, so that $\neg$ appears only immediately in front of predicates, using only De Morgan's laws and $\neg\neg\psi \equiv \psi$.

Because terms include both $\le$ and $\ge$ (standard optimization form can flip from one to another), every negated term can be rewritten as an unnegated term within the same class:

$$\neg(a^\top y \le b) \equiv a^\top y \ge b, \qquad \neg(a^\top y \ge b) \equiv a^\top y \le b.$$

Hence in NNF, $\varphi$ is equivalent to a formula using only $(\wedge, \vee)$ over (unnegated) linear atoms.

By Boolean normalization, $\varphi$ is equivalent to a logical disjunctive normal form: there exists a finite index set $J$ such that

$$\varphi(x,y) \equiv \bigvee_{j \in J} \bigwedge_{\ell \in L_j} \psi_{j\ell}(x,y),$$

where each $\psi_{j\ell}(x,y)$ is a linear inequality in $y$ (either $\le$ or $\ge$).

For each $j \in J$, define the set

$$\mathcal{S}_j(x) := \Big\{\, y \in \mathcal{Y} : \bigwedge_{\ell \in L_j} \psi_{j\ell}(x,y)\,\Big\}.$$

Each $\mathcal{S}_j(x)$ is the intersection of $\mathcal{Y}$ with finitely many halfspaces, hence is a polyhedron. Moreover, any inequality $a^\top y \ge b$ can be rewritten as $(-a)^\top y \le -b$, so $\mathcal{S}_j(x)$ can be expressed in standard form

$$\mathcal{S}_j(x) = \{\, y \in \mathcal{Y} : A_j(x)y \le b_j(x)\,\}.$$

Finally, since $\varphi$ is the disjunction over $j \in J$,

$$\{\, y \in \mathcal{Y} : \varphi(x, y)\,\} = \bigcup_{j \in J} \mathcal{S}_j(x).$$

Setting $\mathcal{C}(x) := \bigcup_{j \in J} \mathcal{S}_j(x)$ yields the claim. $\qquad\square$

### B.3. Proof of Theorem 3.6

**Theorem B.2** (Vertex exactness of the DNF convex-hull projection)**.** *Let $x \in \mathcal{X}$ and $\hat{y} \in \mathbb{R}^d$ be given, and let $w := (y, \eta)$ denote the lifted variables. Assume the lifted rule-feasible set can be written in DNF-union form*

$$\widehat{\mathcal{F}}(x; \hat{y}) = \bigcup_{k \in \mathcal{K}(x)} \widehat{\mathcal{T}}_k(x; \hat{y}), \qquad \widehat{\mathcal{T}}_k(x; \hat{y}) \text{ polyhedral in } w \text{ for each } k, \tag{17}$$

*where $\mathcal{K}(x)$ is a finite index set (e.g., $\mathcal{K}(x) = \prod_{r \in \mathcal{R}(x)} [m_r]$). Let*

$$\widetilde{\mathcal{F}}_{\mathrm{DNF}}(x; \hat{y}) := \mathrm{conv}\big(\widehat{\mathcal{F}}(x; \hat{y})\big).$$

*Consider the DNF projection LP*

$$\min_{w = (y, \eta)} \mathbf{1}^\top \eta \quad s.t. \quad w \in \widetilde{\mathcal{F}}_{\mathrm{DNF}}(x; \hat{y}), \tag{18}$$

*and assume (18) is feasible and bounded.*

*Then every optimal extreme point $w^\star = (y^\star, \eta^\star)$ of (18) satisfies*

$$w^\star \in \widehat{\mathcal{F}}(x; \hat{y}),$$

*i.e., there exists $k^\star \in \mathcal{K}(x)$ such that $w^\star \in \widehat{\mathcal{T}}_{k^\star}(x; \hat{y})$. Consequently, the output component $y^\star$ satisfies all active rule constraints (equivalently, $y^\star \in \mathcal{F}(x)$ after removing lifted/global constraints such as the $\ell_1$ epigraph). In particular, if an LP solver returns an optimal extreme point (e.g., simplex), the returned projection satisfies all rules whenever the original lifted set is nonempty.*

*Proof.* Let $\mathcal{K}^\star \subseteq \mathcal{K}(x)$ index the nonempty polyhedral terms and write each as

$$\widehat{\mathcal{T}}_k(x; \hat{y}) = \{\, w : \widehat{A}_k(x)\, w \le \widehat{b}_k(x; \hat{y})\,\}, \qquad k \in \mathcal{K}^\star,$$

where $w = (y, \eta)$ and $\widehat{A}_k, \widehat{b}_k$ collect all linear inequalities defining the $k$-th lifted term.

By Theorem 2.1 of (Balas, 2018) (disjunctive convex-hull theorem), the convex hull of a finite union of polyhedra admits the extended formulation

$$\mathrm{conv}\Big( \bigcup_{k \in \mathcal{K}^\star} \widehat{\mathcal{T}}_k(x; \hat{y}) \Big) = \left\{ w \;\middle|\; \exists \{w_k\}_{k \in \mathcal{K}^\star}, \{\lambda_k\}_{k \in \mathcal{K}^\star} : \begin{array}{l} \widehat{A}_k(x)\, w_k \le \lambda_k\, \widehat{b}_k(x; \hat{y}) \quad \forall k \in \mathcal{K}^\star, \\ w = \sum_{k \in \mathcal{K}^\star} w_k, \\ \sum_{k \in \mathcal{K}^\star} \lambda_k = 1, \quad \lambda_k \ge 0 \quad \forall k \in \mathcal{K}^\star. \end{array} \right\}. \tag{19}$$

(Our standing assumption that $y \in \mathcal{Y}$ is bounded, together with the copied epigraph/global constraints inside each term, ensures the usual boundedness/closedness conditions under which this polyhedral hull description applies in the present setting.)

Now let $w^\star$ be an *optimal extreme point* of (18), i.e., an optimal vertex of $\widetilde{\mathcal{F}}_{\mathrm{DNF}}(x; \hat{y}) = \mathrm{conv}(\cup_{k \in \mathcal{K}^\star} \widehat{\mathcal{T}}_k)$. Corollary 2.2 of (Balas, 2018) gives the corresponding vertex characterization for the hull: every extreme point of the convex hull of a finite union of polyhedra belongs to (indeed is a vertex of) at least one of the disjunct polyhedra. Therefore, there exists $k^\star \in \mathcal{K}^\star$ such that

$$w^\star \in \widehat{\mathcal{T}}_{k^\star}(x; \hat{y}) \subseteq \widehat{\mathcal{F}}(x; \hat{y}),$$

which proves the first claim.

Finally, membership $w^\star \in \widehat{\mathcal{T}}_{k^\star}(x; \hat{y}) = \bigcap_{r \in \mathcal{R}(x)} \widehat{\mathcal{S}}_{r, k_r^\star}(x; \hat{y})$ implies in particular that $y^\star$ satisfies the unlifted rule consequents in that same selection (dropping epigraph/global constraints), and thus $y^\star \in \bigcap_{r \in \mathcal{R}(x)} \bigcup_{j=1}^{m_r} \mathcal{S}_{rj}(x) = \mathcal{F}(x)$. $\qquad\square$

*Remark* B.3 (On solver outputs). The argument above establishes existence of a rule-satisfying optimal solution. In practice, if the LP is solved with a vertex-returning method (e.g., simplex, or an interior-point method with crossover), the returned primal solution is an extreme point of the feasible polyhedron and therefore satisfies the original rules as above.

## C. Algorithm Pseudocode

---

**Algorithm 1:** DISJUNCTIVENET forward and backward pass

---

**Input:** Input $x$, neural predictor $f_\theta$, rules $\{(\mathcal{A}_r, \mathcal{C}_r)\}_{r=1}^R$, global constraints $\mathcal{G}(x)$, mode mode $\in \{\text{CNF}, \text{DNF}\}$
**Output:** Rule-consistent prediction $y^\star$ and gradients for updating $\theta$
Compute unconstrained prediction $\hat{y} \leftarrow f_\theta(x)$;
Determine active rules $\mathcal{R}(x) \leftarrow \{r \in \{1, \ldots, R\} : x \in \mathcal{A}_r\}$;
**foreach** $r \in \mathcal{R}(x)$ **do**
    **foreach** $j \in \{1, \ldots, m_r\}$ **do**
        Construct lifted disjunct

$$\widehat{S}_{rj}(x; \hat{y}) = \{(y, \eta) : A_{rj}(x)y \leq b_{rj}(x),\ (y, \eta) \in E(\hat{y}),\ y \in \mathcal{G}(x)\}.$$

    Define lifted disjunction $\widehat{C}_r(x; \hat{y}) \leftarrow \bigcup_{j=1}^{m_r} \widehat{S}_{rj}(x; \hat{y})$;
**if** mode $=$ CNF **then**
    Convexify each active rule independently:

$$\widetilde{\mathcal{F}}(x; \hat{y}) \leftarrow \bigcap_{r \in \mathcal{R}(x)} \text{conv}\left(\widehat{C}_r(x; \hat{y})\right).$$

**else**
    Enumerate disjunct selections $\Pi(x) \leftarrow \prod_{r \in \mathcal{R}(x)} [m_r]$;
    Form the DNF convex hull:

$$\widetilde{\mathcal{F}}(x; \hat{y}) \leftarrow \text{conv}\left(\bigcup_{k \in \Pi(x)} \bigcap_{r \in \mathcal{R}(x)} \widehat{S}_{r, k_r}(x; \hat{y})\right).$$

Write $\widetilde{\mathcal{F}}(x; \hat{y})$ as an extended-form LP using convex-combination variables and lifted disjunct copies;
Compute projected prediction:

$$(y^\star, \eta^\star) \in \arg\min_{(y, \eta)} \mathbf{1}^\top \eta \quad \text{s.t.} \quad (y, \eta) \in \widetilde{\mathcal{F}}(x; \hat{y}).$$

Return $y^\star$ as the model prediction;
Backpropagate by implicit differentiation through the KKT conditions of the regularized projection problem:

$$\frac{\partial \mathcal{L}}{\partial \theta} = \frac{\partial \mathcal{L}}{\partial y^\star} \frac{\partial y^\star}{\partial \hat{y}} \frac{\partial f_\theta(x)}{\partial \theta}.$$

---

## D. Examples of lifted projection exactness

### D.1. A one-rule example: exact satisfaction under the lifted convex hull

We first illustrate the projection mechanism with a single active disjunctive rule. Let $y \in [0, 10]$, and suppose the neural network predicts

$$\hat{y} = 4.5.$$

Consider the active rule

$$R: \quad y \leq 3 \ \vee \ y \geq 7.$$

The feasible set is the union of two polyhedra,

$$F = S_1 \cup S_2, \qquad S_1 = \{y : 0 \le y \le 3\}, \qquad S_2 = \{y : 7 \le y \le 10\}.$$

The network prediction $\hat{y} = 4.5$ violates the rule because

$$\hat{y} \notin [0, 3] \cup [7, 10].$$

The projection problem is

$$y^\star \in \arg\min_{y \in F} |y - \hat{y}|.$$

Since $\hat{y} = 4.5$, the closest feasible point is $y = 3$, with distance $1.5$.

To formulate this as a linear program, we introduce the epigraph variable $\eta$, which represents the $\ell_1$ distance to $\hat{y}$. For $\hat{y} = 4.5$, the epigraph constraints are

$$\eta \ge y - 4.5, \qquad \eta \ge 4.5 - y, \qquad \eta \ge 0.$$

We then lift each disjunct into the $(y, \eta)$-space:

$$\widehat{S}_1 = \{(y, \eta) : 0 \le y \le 3,\ \eta \ge y - 4.5,\ \eta \ge 4.5 - y,\ \eta \ge 0\},$$

$$\widehat{S}_2 = \{(y, \eta) : 7 \le y \le 10,\ \eta \ge y - 4.5,\ \eta \ge 4.5 - y,\ \eta \ge 0\}.$$

The lifted projection problem is

$$\min_{y, \eta} \eta \qquad \text{s.t.} \qquad (y, \eta) \in \text{conv}(\widehat{S}_1 \cup \widehat{S}_2).$$

Using the extended convex-hull formulation, we introduce copied variables

$$(y_1, \eta_1), \quad (y_2, \eta_2),$$

and convex-combination weights

$$\lambda_1, \lambda_2 \ge 0, \qquad \lambda_1 + \lambda_2 = 1.$$

The lifted convex-hull LP is

$$
\begin{aligned}
\min \quad & \eta \\
\text{s.t.} \quad & y = y_1 + y_2, \qquad \eta = \eta_1 + \eta_2, \\
& \lambda_1 + \lambda_2 = 1, \qquad \lambda_1, \lambda_2 \ge 0, \\
& 0 \le y_1 \le 3\lambda_1, \\
& 7\lambda_2 \le y_2 \le 10\lambda_2, \\
& \eta_1 \ge y_1 - 4.5\lambda_1, \qquad \eta_1 \ge 4.5\lambda_1 - y_1, \\
& \eta_2 \ge y_2 - 4.5\lambda_2, \qquad \eta_2 \ge 4.5\lambda_2 - y_2, \\
& \eta_1, \eta_2 \ge 0.
\end{aligned}
$$

The optimal extreme-point solution is

$$\lambda_1 = 1, \quad \lambda_2 = 0, \quad y_1 = 3, \quad y_2 = 0, \quad \eta_1 = 1.5, \quad \eta_2 = 0.$$

Therefore,

$$y^\star = y_1 + y_2 = 3, \qquad \eta^\star = \eta_1 + \eta_2 = 1.5.$$

The returned output satisfies the original disjunctive rule exactly:

$$y^\star = 3 \in S_1 \subseteq S_1 \cup S_2 = F.$$

This example shows why the epigraph variable must be included inside the lifted disjuncts before convexification. A naive convexification in the original $y$-space would replace $F = [0, 3] \cup [7, 10]$ by $[0, 10]$, which would incorrectly allow $\hat{y} = 4.5$. In contrast, convexifying the lifted sets $\widehat{S}_1$ and $\widehat{S}_2$ preserves the projection objective, and an optimal extreme point of the lifted hull corresponds to one original feasible disjunct.

### D.2. A two-rule example: exact satisfaction under the DNF lifted hull

We now illustrate the same mechanism with two simultaneously active rules. Let $y \in [0, 10]$, and suppose the neural network predicts

$$\hat{y} = 4.5.$$

Consider two active rules:

$$R_1 : \quad y \leq 3 \ \vee \ y \geq 6, \qquad R_2 : \quad y \leq 4 \ \vee \ y \geq 7.$$

Each rule is a disjunction of two polyhedra:

$$C_1 = S_{11} \cup S_{12}, \qquad S_{11} = \{y : 0 \leq y \leq 3\}, \quad S_{12} = \{y : 6 \leq y \leq 10\},$$

$$C_2 = S_{21} \cup S_{22}, \qquad S_{21} = \{y : 0 \leq y \leq 4\}, \quad S_{22} = \{y : 7 \leq y \leq 10\}.$$

The original feasible set is

$$F = C_1 \cap C_2.$$

The DNF expands the intersection of disjunctions into intersections of disjuncts:

$$F = (S_{11} \cup S_{12}) \cap (S_{21} \cup S_{22})$$
$$= (S_{11} \cap S_{21}) \cup (S_{11} \cap S_{22}) \cup (S_{12} \cap S_{21}) \cup (S_{12} \cap S_{22}).$$

Here,

$$S_{11} \cap S_{21} = [0, 3], \qquad S_{11} \cap S_{22} = \emptyset,$$

$$S_{12} \cap S_{21} = \emptyset, \qquad S_{12} \cap S_{22} = [7, 10].$$

Thus the true feasible set is

$$F = [0, 3] \cup [7, 10].$$

The network prediction $\hat{y} = 4.5$ violates both rules. To project it, we introduce the epigraph variable $\eta$ and lift each nonempty DNF term:

$$\widehat{T}_1 = \{(y, \eta) : 0 \leq y \leq 3, \ \eta \geq y - 4.5, \ \eta \geq 4.5 - y, \ \eta \geq 0\},$$

$$\widehat{T}_2 = \{(y, \eta) : 7 \leq y \leq 10, \ \eta \geq y - 4.5, \ \eta \geq 4.5 - y, \ \eta \geq 0\}.$$

The DNF projection solves

$$\min_{y, \eta} \eta \qquad \text{s.t.} \qquad (y, \eta) \in \text{conv}(\widehat{T}_1 \cup \widehat{T}_2).$$

Using copied variables and convex-combination weights, this is represented as

$$\begin{aligned}
\min \quad & \eta \\
\text{s.t.} \quad & y = y_1 + y_2, \qquad \eta = \eta_1 + \eta_2, \\
& \lambda_1 + \lambda_2 = 1, \qquad \lambda_1, \lambda_2 \geq 0, \\
& 0 \leq y_1 \leq 3\lambda_1, \\
& 7\lambda_2 \leq y_2 \leq 10\lambda_2, \\
& \eta_1 \geq y_1 - 4.5\lambda_1, \qquad \eta_1 \geq 4.5\lambda_1 - y_1, \\
& \eta_2 \geq y_2 - 4.5\lambda_2, \qquad \eta_2 \geq 4.5\lambda_2 - y_2, \\
& \eta_1, \eta_2 \geq 0.
\end{aligned}$$

The closest feasible candidate is $y = 3$, at distance $1.5$ from $\hat{y} = 4.5$. Therefore, the LP has the optimal extreme-point solution

$$\lambda_1 = 1, \quad \lambda_2 = 0, \quad y_1 = 3, \quad y_2 = 0, \quad \eta_1 = 1.5, \quad \eta_2 = 0,$$

which returns

$$y^\star = 3, \qquad \eta^\star = 1.5.$$

This point satisfies both original rules:

$$y^\star = 3 \le 3 \quad \Rightarrow \quad y^\star \in C_1,$$

and

$$y^\star = 3 \le 4 \quad \Rightarrow \quad y^\star \in C_2.$$

Hence,

$$y^\star \in C_1 \cap C_2 = F.$$

This example highlights why DNF gives exact satisfaction for multiple active rules. The DNF first enumerates consistent disjunct selections across all active rules and then takes the lifted convex hull of those feasible intersections. Empty intersections are automatically irrelevant. Because the LP optimum can be chosen as an extreme point of this lifted hull, the returned solution belongs to one original DNF term and therefore satisfies every active rule.

## E. Additional Details and Results

### E.1. Compute environment

All experiments were run on a shared server environment using a single CPU process per run.

**Software and differentiation through optimization.** All models were implemented in `Julia` using `Flux.jl` for neural networks and `DiffOpt.jl` for differentiating through the projection layer (a parametric linear program). We used the `ADAMW` optimizer in Flux for training and fine-tuning. Unless otherwise stated, we did not perform hyper-parameter tuning; we used a single configuration across all methods to ensure controlled comparisons.

### E.2. Synthetic cooling-control problem details.

We consider a simplified cooling-control system with input vector

$$x = (T_a, H, w, \pi, dr, mf, mc),$$

where $T_a$ denotes ambient temperature, $H$ humidity, $w$ workload, $\pi$ electricity price, and $dr$, $mf$, and $mc$ are binary indicators for demand-response events, fan maintenance, and chiller maintenance, respectively. The model predicts continuous control outputs

$$y = (f, c, p) \in [0, 1]^3,$$

corresponding to fan speed, chiller level, and pump power.

System dynamics are captured through simple proxy models. The required cooling load is

$$L(x) = L_0 + L_w w + L_T \max(0, T_a - 20),$$

and feasibility requires $a_f f + a_c c \ge L(x)$. Power consumption is modeled as

$$P(y) = \alpha_f f + \alpha_c c + \alpha_p p,$$

with an always-on constraint $p \ge p_{\min}$. Given reference setpoints $f_{\mathrm{ref}}(w)$ and $c_{\mathrm{ref}}(T_a, w)$, the oracle solution minimizes a weighted objective combining power cost and deviation from these references.

**Rule set.** Domain knowledge is encoded via seven input-dependent logical rules, each activated by a predicate on the context $x$ and enforcing a disjunction of linear constraints on the controls $y = (f, c, p)$. We summarize the rules below; the numerical thresholds come from the fixed parameter configuration used in our experiments (see the accompanying code for the exact values).

- **R1 (hot ambient).** *If* $T_a$ exceeds a hot-temperature threshold, *then*

$$c \ge 0.45 \quad \vee \quad f \ge 0.75.$$

  This captures redundancy under hot conditions: either active cooling (chiller) or high airflow (fan) must increase.

- **R2 (cold ambient).** *If* $T_a$ is below a cold-temperature threshold, *then*

$$c \leq 0.05 \quad \vee \quad f \geq 0.60.$$

This encodes an efficiency heuristic: under cold ambient conditions, heavy chilling is discouraged unless high airflow is needed.

- **R3 (high humidity).** *If* humidity $H$ exceeds a humid threshold, *then*

$$f \leq 0.70 \quad \vee \quad c \geq 0.30.$$

This models a dehumidification trade-off: high airflow is limited unless sufficient cooling is provided.

- **R4 (high workload).** *If* workload $w$ exceeds a high-load threshold, *then*

$$f \geq 0.65 \quad \vee \quad c \geq 0.35.$$

This ensures the controller increases capacity via either actuator under heavy demand.

- **R5 (demand response).** *If* a demand-response event occurs ($\mathrm{dr} = 1$), *then*

$$P(y) \leq P_{\mathrm{dr}} \quad \vee \quad (f \leq 0.50 \wedge c \leq 0.40),$$

where $P(y) = \alpha_f f + \alpha_c c + \alpha_p p$ is the proxy power model. This captures grid constraints: either respect an overall power cap or jointly throttle the main consumers.

- **R6 (fan maintenance).** *If* the fan is under maintenance ($\mathrm{mf} = 1$), *then*

$$f \leq 0.40 \quad \vee \quad c \geq 0.40.$$

This models reduced fan availability: either limit fan usage or compensate via chiller operation.

- **R7 (chiller maintenance).** *If* the chiller is under maintenance ($\mathrm{mc} = 1$), *then*

$$c \leq 0.40 \quad \vee \quad f \geq 0.70.$$

This is symmetric to R6 and models reduced chiller availability: either limit chiller usage or compensate with higher airflow.

**IID vs. OOD evaluation.** We evaluate on two test distributions defined by distinct distributions. Each sample consists of continuous environment/operating variables $(T_a, H, w, \pi)$ and binary event indicators $(\mathrm{dr}, \mathrm{mf}, \mathrm{mc})$. Unless stated otherwise, all draws are independent.

**In-distribution (IID) test set.** The IID test set is drawn from the same distribution as training. Ambient temperature $T_a$ is sampled uniformly from $[0, 40]$ (in °C), relative humidity $H$ is sampled uniformly from $[10, 95]$ (in %), workload $w$ is sampled uniformly from $[0, 1]$, and the price proxy $\pi$ is sampled uniformly from $[0.05, 0.50]$. The event indicators are sampled independently as Bernoulli random variables with

$$\mathrm{dr} \sim \mathrm{Bernoulli}(0.10), \qquad \mathrm{mf} \sim \mathrm{Bernoulli}(0.05), \qquad \mathrm{mc} \sim \mathrm{Bernoulli}(0.05).$$

We generate $n_{\mathrm{test,iid}} = 2000$ IID test points using seed $\texttt{seed\_test\_iid} = 2$.

**Out-of-distribution (OOD) test set.** The OOD test set is drawn from a shifted distribution intended to increase the frequency of extreme operating conditions and events. Specifically, we shift the continuous variables toward hotter and more humid conditions, higher workload, and higher prices:

$$T_a \sim \mathrm{Unif}(18, 45), \quad H \sim \mathrm{Unif}(50, 100), \quad w \sim \mathrm{Unif}(0.4, 1.0), \quad \pi \sim \mathrm{Unif}(0.20, 0.80).$$

We also increase event frequencies:

$$\mathrm{dr} \sim \mathrm{Bernoulli}(0.35), \qquad \mathrm{mf} \sim \mathrm{Bernoulli}(0.08), \qquad \mathrm{mc} \sim \mathrm{Bernoulli}(0.08).$$

We generate $n_{\mathrm{test,ood}} = 2000$ OOD test points using seed $\texttt{seed\_test\_ood} = 3$.

**Neural architecture.** For all learning-based methods in the cooling task, we used a shared MLP backbone with input dimension 7, output dimension 3, and two hidden layers of width 8 with ReLU activations:

$$7 \to 8 \to 8 \to 3.$$

This backbone was used for the base model, penalty model, and all projection variants (CNF, and DNF).

**Training setup.** The base and penalty models were trained for 5 epochs with batch size 256 and learning rate $10^{-3}$ (weight decay 0). Projection-based models were trained with batch size 1 and learning rate $10^{-3}$, using a short training schedule of 1 epoch in this case study (to emphasize the computational overhead and stability properties of the projection layer during optimization). Randomization was controlled through a fixed seed (0), and data shuffling was enabled.

**Penalty and projection configurations.** The penalty baseline used penalty weight $\lambda_{\text{pen}} = 2.0$ and soft-min temperature $\tau_{\text{softmin}} = 10^{-2}$. Projection variants used the same backbone and differed only in the projection operator: CNF projection, DNF projection, and their relaxed counterparts.

### E.3. Computational cost and LP size

We report computational statistics for the synthetic cooling-control task. The projection-based models solve a linear program for each sample, whereas the base and penalty models require only a standard neural-network forward pass. Table 3 reports per-sample inference time. The base and penalty models have negligible overhead, while the CNF and DNF projection layers require tens of milliseconds per sample. The DNF formulation is slightly slower because it constructs a tighter relaxation by expanding intersections of active disjuncts.

*Table 3.* Per-sample inference time on the synthetic cooling task.

| Model | Inference time |
|---|---|
| Base | $< 1\,\mu s$ |
| Penalty | $< 1\,\mu s$ |
| CNF | $25.03\,\text{ms}$ |
| DNF | $28.62\,\text{ms}$ |

Table 4 reports LP sizes for samples with at least one active rule, while Table 5 reports the corresponding statistics for samples with at least two active rules. As expected, DNF produces larger LPs than CNF on average. However, the realized DNF size remains moderate in this benchmark because not all samples activate many rules, and many disjunct intersections are empty or infeasible.

*Table 4.* LP sizes for synthetic cooling samples with at least one active rule.

| Formulation | Vars mean $\pm$ std | Vars min | Vars max | Cons mean $\pm$ std | Cons min | Cons max |
|---|---|---|---|---|---|---|
| CNF | $27.78 \pm 9.75$ | 20 | 62 | $98.28 \pm 39.11$ | 67 | 235 |
| DNF | $32.51 \pm 14.53$ | 23 | 121 | $111.94 \pm 60.27$ | 73 | 485 |

*Table 5.* LP sizes for synthetic cooling samples with at least two active rules.

| Formulation | Vars mean $\pm$ std | Vars min | Vars max | Cons mean $\pm$ std | Cons min | Cons max |
|---|---|---|---|---|---|---|
| CNF | $37.50 \pm 6.54$ | 34 | 62 | $137.28 \pm 26.29$ | 123 | 235 |
| DNF | $44.39 \pm 14.86$ | 37 | 121 | $160.53 \pm 62.69$ | 129 | 485 |

### E.4. Scalability with the number of active rules

The empirical cooling dataset contains at most five active rules per sample. To evaluate scalability beyond the empirical regime, we additionally construct synthetic stress-test LP instances with $k = 6, \dots, 10$ active rules. These rows are not

sampled from the cooling dataset; rather, they are designed to quantify how CNF and DNF scale as the number of candidate disjunctive combinations increases. The DNF size can grow exponentially in $k$ in the worst case, since the number of disjunct combinations grows as $2^k$ in this experiment. In practice, however, the realized DNF size can be substantially smaller than this worst-case bound because many disjunct intersections are empty or infeasible.

*Table 6.* LP size and inference-time scaling with the number of active rules $k$. Counts denote the number of cooling-dataset samples that activate exactly $k$ rules; rows with "–" are synthetic stress-test instances rather than empirical samples.

| $k$ | Formulation | Variables | Constraints | Inference time (s) | Count |
|-----|-------------|-----------|-------------|--------------------|-------|
| 0 | CNF | 9 | 20 | 0.006599 | 57 |
| 0 | DNF | 9 | 20 | 0.009122 | 57 |
| 1 | CNF | 20 | 67 | 0.008361 | 156 |
| 1 | DNF | 23 | 73.1 | 0.007077 | 156 |
| 2 | CNF | 34 | 123.2 | 0.008230 | 139 |
| 2 | DNF | 37 | 129.5 | 0.007697 | 139 |
| 3 | CNF | 48 | 179.3 | 0.008023 | 49 |
| 3 | DNF | 65 | 246.5 | 0.008667 | 49 |
| 4 | CNF | 62 | 235.7 | 0.009787 | 20 |
| 4 | DNF | 121 | 491.0 | 0.010559 | 20 |
| 5 | CNF | 76 | 292.0 | 0.008193 | 1 |
| 5 | DNF | 233 | 997.0 | 0.013100 | 1 |
| 6 | CNF | 87 | 207.0 | 0.005307 | – |
| 6 | DNF | 454 | 1321.0 | 0.019194 | – |
| 7 | CNF | 101 | 241.0 | 0.005601 | – |
| 7 | DNF | 902 | 2761.0 | 0.041093 | – |
| 8 | CNF | 115 | 275.0 | 0.006087 | – |
| 8 | DNF | 1798 | 5769.0 | 0.107464 | – |
| 9 | CNF | 129 | 309.0 | 0.007065 | – |
| 9 | DNF | 3590 | 12041.0 | 0.267479 | – |
| 10 | CNF | 143 | 343.0 | 0.007796 | – |
| 10 | DNF | 7174 | 25097.0 | 0.920017 | – |

Even under this pessimistic stress test, DNF inference remains below one second per sample at $k = 10$, with substantially smaller times for all lower values of $k$. The CNF formulation scales approximately linearly in this setup and remains below 10 ms across all tested values. These results support the practical use of the proposed layer in settings where the number of simultaneously active rules is small to moderate. This regime is also representative of existing neuro-symbolic benchmarks, which typically involve fixed and relatively small rule families.

*Table 7.* Rule counts in representative neuro-symbolic benchmarks. Existing benchmarks typically involve small fixed rule families, comparable to the number of active rules in our experimental settings.

| Benchmark | Rule count |
|-----------|------------|
| MNMath, arithmetic rules | 2 |
| Kand-Logic | 1 |
| CLE4EVR, existential rule | 1 |
| BDD-OIA / SDD-OIA | 1 |
| DeepSeaProbLog, year detection | 5 |
| SATNet / Sudoku / visual Sudoku | 3 |

The purpose of our two benchmark tasks is not only to match the scale of existing neuro-symbolic datasets, but also to evaluate input-dependent mixed-integer constraints with multiple disjunctive rules and multiple linear constraints per disjunct. These features are not represented in the above benchmarks, but arise naturally in scientific and engineering applications.

**Training-time overhead.**   Projection models are more expensive to train than the base and penalty baselines because each forward pass includes an LP solve and each backward pass differentiates through the corresponding optimization layer. In our implementation, the forward pass uses the extended-form LP, while the backward pass uses gradients obtained from a regularized strongly convex approximation, following the differentiable-optimization setup described in Sec. 3.2. The overhead is therefore dominated by the size of the LP relaxation and the number of active rules per sample. This is why the partial-DNF formulation provides a useful trade-off: it can tighten the relaxation by applying basic steps to selected active rules, while avoiding the full combinatorial expansion of DNF.

### E.5. Additional scRNA-seq results

**Neural architecture.**   The base classifier is a two-layer MLP with one hidden layer of width $h = 8$:

$$\texttt{Dense}(n_{\text{in}} \to 8) \; \to \; \texttt{Dense}(8 \to n_{\text{classes}}),$$

initialized with Glorot uniform initialization. For PBMC3k, we have $n_{\text{in}} = 1838$ gene-expression features and $n_{\text{classes}} = 8$ cell types. The same backbone is used across the baselines and projection variants.

**Data splits and training protocol.**   The PBMC3k processed dataset contains 2604 labeled cells, each represented by 1838 gene features and one of 8 classes. We trained with a fixed test fraction of 0.1 and a fixed split seed (42) to keep the test set consistent across methods and dataset-size sweeps. We evaluated dataset-size scaling over training fractions

$$\{0.005, 0.01, 0.05, 0.1, 0.2\},$$

and repeated each experiment for 3 random seeds. Training used batch size 1 and learning rate $3 \times 10^{-3}$.

**Two-stage training for projection models.**   Projection-based models (CNF/DNF and their relaxations) were *fine-tuned* from a base model. Concretely, we first trained the base model for 500 epochs. Then, starting from the base model weights, we fine-tuned each projection model for 15 epochs under its corresponding projection layer and loss. This protocol isolates the effect of the projection layer during training, while keeping the backbone initialization identical across projection variants.

**Baselines.**   We compared against (i) the unprojected base model and (ii) additional constraint-handling baselines (e.g., penalty and finetuned penalty) implemented within the same codebase and trained under the same data splits and evaluation pipeline.

**PBMC test metrics across training set sizes.**   We report mean $\pm$ std across runs for accuracy, macro-F1, macro precision, macro recall, and constraint satisfaction.

| Model | Acc | Macro F1 | Macro Prec | Macro Rec | CSAT |
|---|---|---|---|---|---|
| Base | $0.251 \pm 0.194$ | $0.232 \pm 0.080$ | $0.381 \pm 0.072$ | $0.339 \pm 0.093$ | $0.105 \pm 0.088$ |
| Penalty | $0.468 \pm 0.037$ | $0.339 \pm 0.025$ | $0.385 \pm 0.046$ | $0.466 \pm 0.054$ | $0.046 \pm 0.025$ |
| Finetuned Penalty | $0.254 \pm 0.194$ | $0.224 \pm 0.098$ | $0.389 \pm 0.114$ | $0.328 \pm 0.117$ | $0.180 \pm 0.108$ |
| CNF | $0.329 \pm 0.175$ | $0.324 \pm 0.083$ | $0.459 \pm 0.111$ | $0.460 \pm 0.091$ | $0.709 \pm 0.065$ |
| DNF | $0.342 \pm 0.197$ | $0.326 \pm 0.096$ | $0.450 \pm 0.091$ | $0.462 \pm 0.100$ | $0.908 \pm 0.083$ |
| Rules | $0.180 \pm 0.020$ | $0.162 \pm 0.021$ | $0.181 \pm 0.012$ | $0.315 \pm 0.100$ | $0.980 \pm 0.000$ |

*Table 8.* PBMC test metrics for dataset size $n = 12$ (mean $\pm$ std over runs). CSAT denotes sample-level $\tau$ satisfaction.

| Model | Acc | Macro F1 | Macro Prec | Macro Rec | CSAT |
|---|---|---|---|---|---|
| Base | $0.470 \pm 0.163$ | $0.296 \pm 0.109$ | $0.381 \pm 0.100$ | $0.392 \pm 0.135$ | $0.242 \pm 0.258$ |
| Penalty | $0.431 \pm 0.109$ | $0.319 \pm 0.079$ | $0.361 \pm 0.034$ | $0.411 \pm 0.087$ | $0.052 \pm 0.020$ |
| Finetuned Penalty | $0.414 \pm 0.203$ | $0.293 \pm 0.066$ | $0.465 \pm 0.067$ | $0.358 \pm 0.083$ | $0.317 \pm 0.241$ |
| CNF | $0.503 \pm 0.158$ | $0.407 \pm 0.077$ | $0.496 \pm 0.119$ | $0.539 \pm 0.059$ | $0.804 \pm 0.125$ |
| DNF | $0.501 \pm 0.150$ | $0.402 \pm 0.073$ | $0.488 \pm 0.116$ | $0.534 \pm 0.057$ | $0.951 \pm 0.026$ |
| Rules | $0.167 \pm 0.006$ | $0.158 \pm 0.007$ | $0.161 \pm 0.011$ | $0.343 \pm 0.025$ | $0.980 \pm 0.000$ |

*Table 9.* PBMC test metrics for dataset size $n = 23$ (mean $\pm$ std over runs). CSAT denotes sample-level $\tau$ satisfaction.

| Model | Acc | Macro F1 | Macro Prec | Macro Rec | CSAT |
|---|---|---|---|---|---|
| Base | $0.713 \pm 0.041$ | $0.559 \pm 0.018$ | $0.582 \pm 0.030$ | $0.650 \pm 0.033$ | $0.487 \pm 0.074$ |
| Penalty | $0.753 \pm 0.033$ | $0.567 \pm 0.064$ | $0.600 \pm 0.115$ | $0.632 \pm 0.029$ | $0.464 \pm 0.032$ |
| Finetuned Penalty | $0.673 \pm 0.039$ | $0.538 \pm 0.128$ | $0.574 \pm 0.146$ | $0.599 \pm 0.092$ | $0.611 \pm 0.122$ |
| CNF | $0.726 \pm 0.054$ | $0.588 \pm 0.050$ | $0.614 \pm 0.029$ | $0.703 \pm 0.046$ | $0.846 \pm 0.044$ |
| DNF | $0.723 \pm 0.056$ | $0.584 \pm 0.050$ | $0.610 \pm 0.032$ | $0.699 \pm 0.047$ | $0.948 \pm 0.006$ |
| Rules | $0.193 \pm 0.041$ | $0.175 \pm 0.043$ | $0.196 \pm 0.039$ | $0.362 \pm 0.079$ | $0.980 \pm 0.000$ |

*Table 10.* PBMC test metrics for dataset size $n = 117$ (mean $\pm$ std over runs). CSAT denotes sample-level $\tau$ satisfaction.

| Model | Acc | Macro F1 | Macro Prec | Macro Rec | CSAT |
|---|---|---|---|---|---|
| Base | $0.736 \pm 0.038$ | $0.606 \pm 0.027$ | $0.664 \pm 0.080$ | $0.646 \pm 0.009$ | $0.520 \pm 0.094$ |
| Penalty | $0.753 \pm 0.058$ | $0.528 \pm 0.070$ | $0.553 \pm 0.088$ | $0.556 \pm 0.095$ | $0.575 \pm 0.040$ |
| Finetuned Penalty | $0.686 \pm 0.008$ | $0.551 \pm 0.068$ | $0.617 \pm 0.129$ | $0.591 \pm 0.078$ | $0.654 \pm 0.093$ |
| CNF | $0.691 \pm 0.073$ | $0.566 \pm 0.070$ | $0.607 \pm 0.092$ | $0.658 \pm 0.069$ | $0.863 \pm 0.010$ |
| DNF | $0.691 \pm 0.077$ | $0.549 \pm 0.079$ | $0.590 \pm 0.098$ | $0.645 \pm 0.087$ | $0.918 \pm 0.030$ |
| Rules | $0.193 \pm 0.028$ | $0.167 \pm 0.036$ | $0.184 \pm 0.027$ | $0.311 \pm 0.097$ | $0.980 \pm 0.000$ |

*Table 11.* PBMC test metrics for dataset size $n = 234$ (mean $\pm$ std over runs). CSAT denotes sample-level $\tau$ satisfaction.

| Model | Acc | Macro F1 | Macro Prec | Macro Rec | CSAT |
|---|---|---|---|---|---|
| Base | $0.835 \pm 0.071$ | $0.709 \pm 0.075$ | $0.773 \pm 0.102$ | $0.710 \pm 0.066$ | $0.742 \pm 0.011$ |
| Penalty | $0.829 \pm 0.035$ | $0.675 \pm 0.164$ | $0.720 \pm 0.170$ | $0.680 \pm 0.175$ | $0.595 \pm 0.102$ |
| Finetuned Penalty | $0.776 \pm 0.068$ | $0.648 \pm 0.094$ | $0.662 \pm 0.109$ | $0.709 \pm 0.050$ | $0.778 \pm 0.067$ |
| CNF | $0.804 \pm 0.055$ | $0.643 \pm 0.080$ | $0.646 \pm 0.084$ | $0.736 \pm 0.055$ | $0.856 \pm 0.006$ |
| DNF | $0.813 \pm 0.056$ | $0.666 \pm 0.081$ | $0.666 \pm 0.094$ | $0.756 \pm 0.055$ | $0.882 \pm 0.020$ |
| Rules | $0.189 \pm 0.021$ | $0.167 \pm 0.014$ | $0.181 \pm 0.022$ | $0.337 \pm 0.031$ | $0.980 \pm 0.000$ |

*Table 12.* PBMC test metrics for dataset size $n = 469$ (mean $\pm$ std over runs). CSAT denotes sample-level $\tau$ satisfaction.

