# OpenReview forum: "DisjunctiveNet: Neural Symbolic Learning via Differentiable Convexified Optimization Layers"
_ICML.cc/2026/Conference — ICML 2026 regular_

### Official Review · Reviewer_qW2C · 2026-03-11

**Soundness:** 1
**Presentation:** 1
**Significance:** 2
**Originality:** 3
**Overall Recommendation:** 3
**Confidence:** 3

**Summary:**

Disjunctive Net is an approach to integrate domain knowledge formulated non-convex constraints into the neural learning. It is quite general, supporting convex and non-convex constraints formed by logical statement over linear inequalities. It can even handle input-dependent constraints and mixed-integer domains. It approaches this problem by deriving a hierarchical convex relaxation, a convex over-approximation of the non-convex constrained space. As the non-convex regions can be disconnected, the convex over-approximation can be arbitrarily bad (compare figure 1). The over-approximation sits in its looseness between a the tighter but expensive convex hull obtained via the DNF and the loose but cheap convex hull relaxation via the CNF. The paper starts off with a solid theoretical analysis but follows through with a rather shallow experimental section, most importantly a clear description of the algorithm in pseudocode, including a discussion of the complexities of each step, vectorization and GPU-friendliness is missing.

**Compliance With Llm Reviewing Policy:**

Affirmed.

**Final Justification:**

The paper has merit but some issues are holding it back. First off, even after the reply to my rebuttal acknowledgement I feel like the experimental section is a weak-spot. If the paper proposed new benchmarks, they need to be clearly motivated to judge impact, especially if the claim is that "In real applications, such rules are common, but current benchmarks do not yet reflect this setting". Also, the presentation of the original submission is poor and needs a major revision, for example the pseudo-code is not given at all and the  concise pseudocode in the rebuttal is (probably due to space constraints) very concise. With the paper proposing a new algorithm it should be detailed in full in the paper, otherwise it is very easy to miss important details/choice during reading the prose.

**Key Questions For Authors:**

Q1: How do the experimental results change like with per-method hyper parameter optimization?

Q2: What are all the steps needed to compute the algorithm. Can they be run on the GPU? Can they be vectorized?

Q3: What is the computational overhead of the method? How does it empirically scale with dimensions, number of polyhedral sets etc.?

Q4: It appears we assume boundedness of our feasible set, is this right?

Q5: The convex hull relaxation is motivated on page 4 (lower left line 211) by allowing differentiability. But it is unclear why this is? Especially as non-convex constraints were integrated into neural networks in the input-independent setting [1] via projections.




[1] Beyond the convexity assumption: Realistic tabular data generation under quantifier-free real linear constraints, Stoian et. al, 2025

**Limitations:**

yes

**Strengths And Weaknesses:**

Soundness is a major issue with the paper. The paper first starts off with a solid theoretical analysis but lacks in the parts following after it: a clear presentation of the resulting algorithm and experimental evaluation. For example, the authors write that all methods were evaluated under a fixed configuration without per-method hyper parameter evaluation. I do not think this is a reasonable assumption as the different methods might require different hyper parameter values and some might be more suitable for some method or the other. Also, the evaluation is only given in accuracy and constraint satisfaction, but timings are never provided. Also, the experiments seem to be non-standard.

Presentation wise, the paper is well written but significantly lacks clarity. A first major complaint is that we never have a clear presentation of the actual algorithm in pseudocode with stated assumptions on the inputs, computational complexity etc. This makes it hard to talk about what the paper is about, as it proposes an algorithm but never formally defines it. Another major presentation wise-complaint is that the paper should make a clearer distinction between the original, non-convex constraints and the convex hull. E.g. it seems in the abstract the convex relaxation enable exact rule satisfaction but a convex relaxation cannot enable exact rule satisfaction: it is an over-approximation of the original constraint set. We loose the guarantees. This confusion would also be clearer if we arrive at a succinct, clear, presentation of the resulting algorithm. Writing wise, the complaints are only minor, e.g. figure 1 is hard to parse and some assumptions are only stated later in the paper, for example it seems to be that the constrained space is assumed to be bounded and should be clearly state in section 3.1.

Significance wise, the paper appears to be motivated but not convincing. If we e.g. have constraints poking holes in our constraint sets (via the statement that the variable x does **not** belong to some set), these holes get fully covered by the convex hull but they might have been vital. Is this now a major limitation or not? After reading the paper the question is quite undressed. The pure experimental advantages compared to baselines are also hard to judge as the experimental section is limited.

The paper is original and provides a nice insight into different convex relaxation of the non convex constraint set. This might be useful for researchers working on other problems and needing different kinds of convex relaxations. The approach appears novel.

---

> ### Author Rebuttal · Authors · 2026-03-31
>
> We thank the reviewer for the careful reading and for recognizing the originality of the convexification perspective.
>
> **Comment on Weaknesses:**
>
> 1. **Tuning and Scalability (Q1, Q3):** We agree that using a shared untuned configuration is not fair. To address this, we have now run additional experiments over learning rate [0.0001, 0.0003, 0.001, 0.003] and penalty weights [0.01, 0.1, 1, 10, 100].
>
> **Tuning results**
>
> *base* = unconstrained NN, *pen* = penalty from scratch, *fine pen* = penalty finetuned from a pretrained base model, agg pen = penalty with a large penalty weight. Learning rate is excluded as the best results were observed with 0.001 uniformly.
>
> **Cooling task** (100 samples)
>
> |Model|$\lambda$| IID MSE|IID CSAT|OOD MSE|OOD CSAT|
> |---|---|---|---|---|---|
> |base|-|0.00986|0.537|0.01284|0.329
> |pen|0.1|0.00611|0.783|0.01183|0.492
> |fine pen|1|0.00609|0.790|0.01050|0.460
> |agg pen|100|0.04560|0.928|0.02791|0.815
> |DNF|-|0.00460|1.0|0.02708|1.0|
>
> **scRNA task** (12 samples)
>
> |Model|$\lambda$|Train CSAT|Test CSAT|F1|
> |---|---|---|---|---|
> |Base|–|0.833|0.069|0.143|
> |Penalty|1|0.944|0.131|0.360|
> |Fine pen|1|0.944|0.157|0.243|
> |Agg pen|10|1.000|0.046|0.339|
> |DNF|–|1.000|1.000|0.358|
>
> These results show that tuning helps, but a clear MSE-CSAT tradeoff is seen. Large penalty weights show high training satisfaction, but at substantial cost in MSE, and with poor transfer to test satisfaction. DNF maintains perfect rule satisfaction while remaining competitive or better on MSE.
>
> **Computational cost**
>
> We provide results only for the cooling task in the rebuttal, to keep it within limits.
>
> **Inference Time**
> | Model | Inf Time|
> |---|---|
> |Base|<1 $\mu$s|
> |Penalty|<1 $\mu$s|
> |CNF|25.03 ms|
> |DNF|28.62 ms|
>
> CNF and DNF require 25.03 ms and 28.62 ms per sample, respectively, so exact projection remains lightweight at this scale, with only a modest overhead for DNF. Efficiency improvements using parallelized threads are an important future direction.
>
> **Problem size**
>
> At least 2 active rules
> |Mode|Mean vars|Max vars|Mean cons|Max cons|
> |---|---|---|---|---|
> |CNF|37.50 ± 6.54|62|137.28 ± 26.29|235|
> |DNF|44.39 ± 14.86|121|160.53 ± 62.69|485|
>
> DNF is understandably larger on average. In practice, the realized DNF size can often be smaller than the worst-case exponential bound, as several disjuncts might not have feasible intersections.
>
> 2. **Pseudocode and Assumptions (Q2, Q4):** The current paper does assume bounded output domains in Theorems 3.2 and 3.6. We will add the assumption in the main text explicitly.
>
> Regarding vectorizability, the work is not a batched GPU implementation. The projection layer is presently CPU-based because we need a vertex-returning LP method. These are not optimized for GPUs as they are sequential, perform sparse memory access, etc. Scalable GPU optimization for LP is not mature yet and has only been developed for first-order methods, like PDHG, that do not have extreme point guarantees. The most realistic near-term implementation is a hybrid: NN on GPU and LP on parallel CPU threads.
>
> A concise pseudocode
>
> *Forward* : (i) Unconstrained prediction: $\hat{y} = f_{\theta}(x)$ (ii) Determine active rules $F(x)$ (Eq. 2) (iii) Build the feasible set and convexify $\hat{F}(x; \hat{y})$ (Eqs. 8, 10, 12, 13) (iv) Solve the projection problem (Eq. 9)
> *Backpropagation*: (v) differentiate through the LP layer using the implicit function theorem over KKT conditions.
>
>
> 3. **Hole / negation/differentiable**
>
> Our exactness claim is **not** that convexification preserves the original nonconvex geometry pointwise. Instead, convexification is applied to the *DNF lifted form*, where the epigraph variable $\eta$ is introduced (Eqs. 9–10) to represent the $l_1$ distance in LP form. $\eta$ **must be** included in the convexification scheme. Theorem 3.6 provides the exactness result.
>
> To illustrate, consider a “hole” (negation). Let $y \in (0,8)$ with the rule $y \notin (4,6)$, creating a nonconvex set. This can be written as a union of two polytopes:
> - $P_1 = \{ y \mid 0 \le y \le 4 \}$
> - $P_2 = \{ y \mid 6 \le y \le 8 \}$
>
> Suppose $\hat{y} = 4.5$, which lies in the convex hull of $P_1 \cup P_2$ but not in $P_1 \cup P_2$.
>
> The lifted convex hull formulation (Eq. 9) considers the convex hull of:
> - $\hat{P}_1 = \{ (\eta, y) \mid 0 \le y \le 4,\ \eta \ge y - 4.5,\ \eta \ge 4.5 - y \}$
> - $\hat{P}_2 = \{ (\eta, y) \mid 6 \le y \le 8,\ \eta \ge y - 4.5,\ \eta \ge 4.5 - y \}$
>
> The extreme points of the *lifted convex hull* are $(\eta, y) = (0.5, 4)$ and $(1.5, 6)$.
> Clearly, $(0.5, 4)$ is the optimal extreme point. The convexification to LP allows differentiability while keeping exactness.
>
> 4. The cited work Stoian et. al, 2025 is a good reference to add. However, it addresses a single linear inequality inside each disjunct. Our framework allows multiple linear inequalities in *conjunction* within each disjunct. Extending their method to multiple linear constraints in each disjunct is not trivial.

---

> > ### Author Rebuttal · Reviewer_qW2C · 2026-04-02
> >
> > I want to thank the authors for engaging with my points. Addressing the question is a clear step forward in my opinion, especially addressing Q1. Nethertheless, I want to keep my score, due to e.g. the experiments seem to be non-standard, presentation of the submission being poor and the computational challenges as it seems so CPU dependent.

---

> > > ### Author Response · Authors · 2026-04-05
> > >
> > > We thank the reviewer again for the follow-up. We understand that the remaining concerns are mainly about experimental standardization, presentation clarity, and GPU compatibility, rather than the core theorem itself.
> > >
> > > 1. **Non standard experiments**
> > >
> > > We agree that our work does not use the standard benchmarks most commonly seen in neuro-symbolic learning. The main reason is that these benchmarks almost always use a fixed, small rule family over discrete concepts (see the following table). These are all valuable, but they do **not** capture the regime we target: **hard, input-dependent mixed logical-linear constraints**. Especially, the proposed approach can satisfy **an arbitrary number of linear inequalities in each disjunct** simultaneously. In real applications, such rules are common, but current benchmarks do not yet reflect this setting. This is why we created a synthetic control task and an scRNA benchmark that can help highlight this distinction. We hope the contribution of two new datasets can motivate more research in this problem setting.
> > >
> > > | Benchmark                                  | Rule Count |
> > > |--------------------------------------------|------------|
> > > | MNMath (arithmetic rules) [1]              | 2          |
> > > | Kand-Logic (logic) [1]                     | 1          |
> > > | CLE4EVR (existential rule) [1]             | 1          |
> > > | BDD-OIA / SDD-OIA [1]                      | 1          |
> > > | DeepSeaProbLog year detection [2]          | 5          |
> > > | SATNet / Sudoku / visual Sudoku [2]        | 3          |
> > > | MultiplexNet CIFAR100 [4]                  | 20         |
> > >
> > >
> > >
> > > 2. **Presentation**
> > >
> > > We agree the presentation can be improved. In retrospect, part of the challenge is that the paper sits between two communities with very different conventions and terminologies: differentiable optimization layers [5,6,7] and neuro-symbolic learning. With the excellent reviews we have received, we are now in a much better position to present the material more clearly in revision. As suggested by the reviewer, we will add the pseudocode and a worked-out toy example in a figure in the revision, ensuring easier digestibility for readers from both communities. We would appreciate any further specific clarification questions from reviewer to help improve the paper if the reviewer still find the presentation is unclear.
> > >
> > >
> > > 3. **On CPU dependence**
> > >
> > > Although our framework is currently CPU-based, we would like to emphasize that this is not unique to our work. CPU solver-based differentiable optimization layers are common in highly cited ICML and NeurIPS papers, such as OptNet [5], Differentiable Convex Optimization Layers [6], and Differentiable Distributionally Robust Optimization Layers [7]. In our case, the projection layer remains CPU-based because existing solvers are already highly efficient on CPUs, and our target applications are in *low data regime* that involve small to moderate numbers of active constraints rather than large-scale batched optimization.
> > >
> > > GPU-based LP solvers have only emerged in the past few years [8], and they are not yet competitive with CPU-based solvers for general linear programs. The reason is that most LP algorithms require frequent matrix inverse calculations that are not amenable to run on GPUs.
> > >
> > > However, if we take a step back and say we want to solve our problems fully on GPU, since our framework is implemented in the general optimization modeling language JuMP.jl, switching to a GPU-based solver such as that in [8] requires only one line of code change (instead of setting the solver to Gurobi, setting the solver to cuPDLP.jl [8]). Given that GPU-based LP solvers are still relatively immature, we do not see a strong justification for adopting them at this stage. Developing a competitive GPU-based LP solver is itself a broader research area that lies beyond the scope of this work.
> > >
> > > Our primary contribution is that, to the best of our knowledge, this is the first framework that enables differentiable enforcement of MILP-representable constraints.
> > >
> > >
> > > [1] Bortolotti et al. A Neuro-Symbolic Benchmark Suite for Concept Quality and Reasoning Shortcuts. NeurIPS 2024 Datasets and Benchmarks Track.
> > >
> > > [2] De Smet et al. Neural Probabilistic Logic Programming in Discrete-Continuous Domains (DeepSeaProbLog). UAI 2023 / ICLR 2023
> > >
> > > [3] Wang et al. SATNet: Bridging Deep Learning and Logical Reasoning Using a Differentiable Satisfiability Solver. ICML 2019.
> > >
> > > [4] Hoernle et al. MultiplexNet: Towards Fully Satisfied Logical Constraints in Neural Networks. AAAI 2022
> > >
> > > [5] Amos and Kolter. OptNet: Differentiable Optimization as a Layer in Neural Networks. ICML 2017.
> > >
> > > [6] Agrawal et al. Differentiable Convex Optimization Layers. NeurIPS 2019.
> > >
> > > [7] Ma et al. Differentiable Distributionally Robust Optimization Layers. ICML 2024.
> > >
> > > [8] Lu and Yang. cuPDLP.jl: A GPU implementation of restarted primal-dual hybrid gradient for linear programming in Julia. Operations Research, 2025

---

### Official Review · Reviewer_6WGp · 2026-03-12

**Soundness:** 3
**Presentation:** 3
**Significance:** 2
**Originality:** 3
**Overall Recommendation:** 4
**Confidence:** 3

**Summary:**

This paper proposes a framework for strict enforcement of input-dependent logical rules over neural network outputs, where each rule is a finite disjunction of linear constraints. This representation is as expressive as mixed-integer linear programming as well as quantifier-free linear real arithmetic. The key to making this architecture differentiable is through convexifying the disjunctive constraints and representing the resulting DNF as a differentiable linear program. Results from the evaluation on synthetic cooling control and real-world RNA sequencing highlight the approach’s data-efficiency and adherence to strict rules, compared to the baselines.

**Compliance With Llm Reviewing Policy:**

Affirmed.

**Final Justification:**

I remain optimistic about this work due to the stated strengths and clarifications from the rebuttal, so I maintain my original score. I hope that the additional results can be incorporated into the revised manuscript.

**Key Questions For Authors:**

- Would it be possible to train the NN from scratch, i.e., in the absence of intermediate labels like fan speed, chiller usage, etc. (as in the cooling task)?
- I’d appreciate some clarification on the input-dependent logical rules and why other logic programming frameworks can’t encode them. Specific examples of if-then rules from the benchmarks would be helpful here.
- Related to the question above, but can’t these logic programming approaches be used as baselines, like DeepProbLog or DeepSeaProbLog [2], the latter of which can operate over continuous probability distributions which would handle the setting in the cooling-control problem?
- Are there any ideas for how to choose which rules to intersect at each iteration of convexification, rather than applying a fixed sequence of steps?

[2] Neural Probabilistic Logic Programming in Discrete-Continuous Domains, UAI 2023

**Limitations:**

yes

**Strengths And Weaknesses:**

**Strengths**
- The convex hull relaxation for ensuring differentiability of the constraints is a really nice idea. The evaluation includes CNF/DNF comparison to understand the tradeoffs of the different projection layers.
- I really like that there is a real-world benchmark here with RNA sequencing, especially when most neurosymbolic benchmarks in the literature seem to be synthetic.
- The method demonstrates much lower MSE and higher satisfaction than the baselines over all dataset sizes. The difference is especially pronounced for smaller datasets, demonstrating data efficiency.

**Weaknesses**
- This paper proposes using a base trained NN to train the projection layer (where I assume you freeze the NN parameters when training the projection layer). This is unlike most other neurosymbolic learning literature which trains the NN from scratch using only the end labels, e.g., DeepProbLog and ABL [1]. But maybe this makes sense considering using only a yes/no constraint satisfaction label would make training the NN from scratch challenging.
- Maybe I am misunderstanding what is meant by “input-dependent logical rules”, but I think that neurosymbolic frameworks that use probabilistic logic programs can handle these “if-then” rules. For example, DeepProbLog can encode implication. Isn’t this an input-dependent rule?
- There is discussion of difference in computational complexity between CNF and DNF approaches but experiments do not demonstrate this difference empirically.

[1] Bridging Machine Learning and Logical Reasoning by Abductive Learning, NeurIPS 2019

---

> ### Author Rebuttal · Authors · 2026-03-31
>
> We thank the reviewer for the positive assessment, especially the convex-hull idea, the CNF/DNF comparison, and the inclusion of a real-world scRNA benchmark. We are also encouraged that the reviewer found the low-data gains and rule-satisfaction results meaningful.
>
> 1. **Training Protocol (Q1):** The projection layer itself does not add any trainable parameters. We agree that our use of ‘finetuned’ may be confusing, but the NN parameters are **not frozen**. We will revise the presentation. As our framework is model-agnostic, it is possible to freeze when large NN backbones are involved.
> The two-stage training was a design choice as it (i) reduces computational overhead as the base model is faster, (ii) allows the model to first learn a good unconstrained approximation before hard rule enforcement is introduced. It has been observed in other work [1] (Appendix C.1) that enforcing constraints initially could cause unstable training and suboptimal performance.
>
> 2. **Input-dependent rules, other methods (Q2, Q3):** We agree that implication-like rules can be represented in probabilistic logic frameworks such as DeepSeaProbLog. The contribution of our framework is not the ability to express “if-then” rules, but to **strictly** enforce them (Theorem 3.6). Existing neuro-symbolic methods (Table 1, Appendix A), to the best of our knowledge, do not simultaneously support (i) input-dependent rules, (ii) mixed logical-linear constraints, (iii) 100\% rule satisfaction
> A concrete example is: if ambient temperature (T) is high (input only), then $c\geq0.2\vee f\geq0.7$, where chiller ($c$) and fan ($f$) speeds are outputs.
>
> 3. **CNF-DNF tradeoff:** We agree that computational overhead needs to be illustrated empirically as well. We have performed additional experiments, which we will add to the revision.
>
> **Computational Overhead Results**
>
> We provide results only for the cooling task in the rebuttal, to keep it streamlined and short. scRNA results will be included in the revised text.
>
> **Inference Time**
> | Model | Inf Time per sample |
> |---|---|
> |Base|<1 $\mu$s|
> |Penalty|<1 $\mu$s|
> |CNF|25.03 ms|
> |DNF|28.62 ms|
>
> For the cooling task, CNF and DNF require 25.03 ms and 28.62 ms per sample, respectively, so exact projection remains lightweight at this scale, with only a modest overhead for DNF. Efficiency improvements using parallelized thread computation are an important future direction for our work.
>
> **Problem size**
>
> Samples with at least 1 active rule
> | Formulation | Vars (mean ± std) | Vars min | Vars max | Cons (mean ± std) | Cons min | Cons max |
> | --- | --- | --- | --- | --- | --- | --- |
> | CNF | 27.78 ± 9.75 | 20 | 62 | 98.28 ± 39.11 | 67 | 235 |
> | DNF | 32.51 ± 14.53 | 23 | 121 | 111.94 ± 60.27 | 73 | 485 |
>
> Samples with at least 2 active rules
> | Formulation | Vars (mean ± std) | Vars min | Vars max | Cons (mean ± std) | Cons min | Cons max |
> | --- | --- | --- | --- | --- | --- | --- |
> |CNF | 37.50 ± 6.54 | 34 | 62 | 137.28 ± 26.29 | 123 | 235 |
> |DNF | 44.39 ± 14.86 | 37 | 121 | 160.53 ± 62.69 | 129 | 485 |
>
> Problem sizes are consistent with the expected CNF-DNF tradeoff. DNF is larger on average, and the gap widens once multiple rules are active. This is expected, since with only one active rule, the CNF and DNF formulations are the same. In practice, the realized DNF size can often be much smaller than the worst-case exponential bound, as several disjuncts might not have feasible intersections, especially with global constraints involved.
>
> 4. **Q4:** We appreciate this important question about optimal rule-ordering for sequential convexification. There is currently no well-defined notion of an “optimal” sequence. As our main focus for this work was to illustrate the utility of CNF and DNF, we assumed a lexicographic ordering. At this stage, considering the complexity of this task, for this paper, we can only provide some heuristics
>
> **(i) $\lambda$ fractionality-based heuristic.** We can first solve the *CNF* relaxation and inspect the corresponding $\lambda$-values. If a rule already has a near-integral $\lambda$, then the CNF relaxation is already tight. By contrast, rules with highly fractional values indicate the resulting projection lies between multiple disjuncts. A reasonable heuristic is therefore to keep near-integral rules in CNF form and prioritize the most fractional rules for DNF expansion for a tighter relaxation.
>
> **(ii) Interaction-based heuristic.**  Another heuristic is to prioritize rules that interact most strongly with other active rules. For example, one could rank rules according to how many variables they share with other rules, or how often they are simultaneously active. Convexifying an isolated rule may have a limited impact, whereas convexifying a rule that is heavily coupled with others can tighten the feasible region much more substantially.
>
> [1] Youngjae Min and Navid Azizan (2024). Hardnet: Hard-constrained neural networks with universal approximation guarantees.

---

> > ### Author Rebuttal · Reviewer_6WGp · 2026-04-04
> >
> > Thank you for the rebuttal and addressing my concerns. I remain optimistic about the work, and I hope that some of these additional results can be incorporated in the revision. I maintain my score.

---

### Official Review · Reviewer_2w9W · 2026-03-12

**Soundness:** 3
**Presentation:** 3
**Significance:** 2
**Originality:** 2
**Overall Recommendation:** 4
**Confidence:** 4

**Summary:**

This paper proposes DisjunctiveNet, a framework for enforcing input-dependent logical and MILP-representable constraints on neural network outputs via differentiable projection layers. The key idea is to represent rules as unions of polyhedra and then apply CNF/DNF-style convex-hull reformulations so that projection can be implemented as an LP layer. The problem is meaningful, the formulation is technically interesting, and the CNF-to-DNF tightening story is intuitive and well illustrated.

**Compliance With Llm Reviewing Policy:**

Affirmed.

**Final Justification:**

My concern has been resolved. I encourage the authors to add more of the empirical evidence provided here, especially the tuned baselines and inference time analysis, as well as the dedicated Limitations section.

**Key Questions For Authors:**

1. In Eq. (15), should the aggregation be $\sum_k y_k = y$ rather than $\sum_k y_k = \hat{y}$? Please state exactly what was implemented in the forward pass.
2. Which LP solver / algorithm was used in the forward pass, and does it return extreme-point solutions, or use crossover to a vertex solution? This matters directly for the practical meaning of the exactness claim.
3. Did you try tuned penalty baselines or the same pretraining-plus-finetuning protocol for the penalty model?
4. What fraction of PBMC3k samples had contradictory active rules, and how would CSAT change if those samples were included in the denominator?

**Limitations:**

Limitations are not discussed

**Strengths And Weaknesses:**

# Strengths

The paper addresses an important problem: hard enforcement of input-dependent logical constraints inside end-to-end neural models, which is relevant in low-data and scientific settings. The formulation is coherent, and the CNF/DNF convexification story is intuitive and reasonably well supported by the geometric presentation and the sequential-convexification experiments.

# Weaknesses

1. In Eq. (15), the paper writes the DNF aggregation as $\sum_k y_k = \hat{y}$, which appears inconsistent with the projection objective, where $\hat{y}$ is the unconstrained NN output and $y$ is the projected decision variable. This may be a typo, but it appears in the main method description and materially affects confidence in the presentation.
2. Theorem 3.6 and Remark 3.7 only guarantee satisfaction for extreme-point LP solutions, i.e., under a solver-side condition, whereas the abstract and conclusion are written more like unconditional exact guarantees.
3. Projection models are initialized from a trained base model and then fine-tuned, while all methods are evaluated without per-method hyperparameter tuning. This especially weakens the penalty baseline and makes the projection-vs-penalty comparison less fair than the paper suggests.
4. The paper repeatedly frames CNF/DNF as a tightness-versus-scalability trade-off, and Remark 3.4 explicitly notes exponential growth of DNF terms, but the experiments report essentially no computational evidence for that trade-off such as runtime, memory, or LP size. As written, only the tightness side is demonstrated.
5. The scRNA CSAT numbers should be interpreted more carefully: contradictory-rule samples bypass projection and are excluded from CSAT, so the reported 100% DNF satisfaction is only on the subset of non-contradictory cases. The paper should report the frequency of such excluded samples.

---

> ### Author Rebuttal · Authors · 2026-03-31
>
> We thank the reviewer for the careful reading and for recognizing the soundness of the work. We have answered the questions and commented on the weakness in unison to save space.
>
> 1. **Eq (15) (Q1):** The reviewer is correct. The aggregation should be the projected variable $y^*$, not the unconstrained NN output $\hat y$. We will correct this typo. This is a presentation error only; the underlying formulation and implementation are unchanged.
>
> 2. **Exactness claim (Q2):** We will revise the abstract and conclusion to state the solver conditions (Theorem 3.6, Remark 3.7) precisely. Our forward-pass solver uses Gurobi’s concurrent optimizer, with dual simplex and barrier-with-crossover enabled, which guarantees an extreme point solution. In addition, an extreme point solution is guaranteed for most commercial and open-source LP solvers.
>
> 3. **Penalty baseline (Q3):** We agree that this is a fair concern. To address this, we ran an additional matched comparison in which the penalty baseline is also finetuned from the pretrained base model, and tuned all methods over learning rate [0.0001, 0.0003, 0.001, 0.003] and penalty weights [0.01, 0.1, 1, 10, 100].
>
> **Hyper-parameter tuning results**
>
> *base* = unconstrained NN, *pen* = penalty from scratch, *fine pen* = penalty finetuned from a pretrained base model, agg pen = penalty with a large penalty weight. Learning rate is excluded as the best results were observed with 0.001 uniformly.
>
> **Cooling task** (100 samples)
>
> |Model|$\lambda$| IID MSE|IID CSAT|OOD MSE|OOD CSAT|
> |---|---|---|---|---|---|
> |base|-|0.00986|0.537|0.01284|0.329
> |pen|0.1|0.00611|0.783|0.01183|0.492
> |fine pen|1|0.00609|0.790|0.01050|0.460
> |agg pen|100|0.04560|0.928|0.02791|0.815
> |DNF|-|0.00460|1.0|0.02708|1.0|
>
> **scRNA task** (12 samples)
>
> |Model|$\lambda$|Train CSAT|Test CSAT|F1|
> |---|---|---|---|---|
> |Base|–|0.833|0.069|0.143|
> |Penalty|1|0.944|0.131|0.360|
> |Fine pen|1|0.944|0.157|0.243|
> |Agg pen|10|1.000|0.046|0.339|
> |DNF|–|1.000|1.000|0.358|
>
> These results show that tuning improves the penalty baseline, but it still exhibits a clear performance-versus-constraint-satisfaction tradeoff. Very large penalty weights can approach high training satisfaction, but at substantial cost in predictive performance and with poor transfer to test satisfaction, especially in low-data settings. By contrast, DNF maintains perfect rule satisfaction while remaining competitive or better on predictive metrics.
>
> 4. **Tightness vs Scalability:** We agree that the paper demonstrates the tightness side of the CNF-DNF tradeoff more explicitly than the computational side. We will add inference time and LP-size for an empirical perspective.
>
> **Computational Overhead Results**
>
> We provide results only for the cooling task in the rebuttal, to keep it within char limits. scRNA results will be included in the revised text.
>
> **Inference Time**
> |Model|Inf Time per sample|
> |---|---|
> |Base|<1 $\mu$s|
> |Penalty|<1 $\mu$s|
> |CNF|25.03 ms|
> |DNF|28.62 ms|
>
> For the cooling task, CNF and DNF require 25.03 ms and 28.62 ms per sample, respectively, so exact projection remains lightweight at this scale, with only a modest overhead for DNF. Efficiency improvements using parallelized thread computation are an important future direction for our work.
>
> **Problem size**
>
> Samples with at least 1 active rule
> | Formulation | Vars (mean ± std) | Vars min | Vars max | Cons (mean ± std) | Cons min | Cons max |
> | --- | --- | --- | --- | --- | --- | --- |
> | CNF | 27.78 ± 9.75 | 20 | 62 | 98.28 ± 39.11 | 67 | 235 |
> | DNF | 32.51 ± 14.53 | 23 | 121 | 111.94 ± 60.27 | 73 | 485 |
>
> Samples with at least 2 active rules
> | Formulation | Vars (mean ± std) | Vars min | Vars max | Cons (mean ± std) | Cons min | Cons max |
> | --- | --- | --- | --- | --- | --- | --- |
> |CNF | 37.50 ± 6.54 | 34 | 62 | 137.28 ± 26.29 | 123 | 235 |
> |DNF | 44.39 ± 14.86 | 37 | 121 | 160.53 ± 62.69 | 129 | 485 |
>
> Problem sizes are consistent with the expected CNF-DNF tradeoff. DNF is larger on average, and the gap widens once multiple rules are active. This is expected, since with only one active rule, the CNF and DNF formulations are the same. In practice, the realized DNF size can often be much smaller than the worst-case exponential bound, as several disjuncts might not have feasible intersections, especially with global constraints involved.
>
> 5. **Contradictory samples (Q4):** There was 1 contradictory sample in the PBMC3k test set (<1%). We excluded it because satisfying all active rules is impossible for any method. Including that sample would reduce DNF rule satisfaction from 100.0% to about 99.1%. Addressing such samples is a focus for robustness or adversarial research and out of scope for this work.
>
> **Limitations.** We also agree that the paper should include an explicit limitations paragraph. We will add that:
>  (i) DNF can be exponentially larger than CNF in the number of active rules,
>  (ii) the exact DNF satisfaction result is solver-conditional as above

---

> > ### Author Rebuttal · Reviewer_2w9W · 2026-04-04
> >
> > Thank you for your rebuttal. The rebuttal addresses all of my questions and improves the paper in several aspects, but some core concerns remain only partially resolved.
> >
> > The Eq. (15) issue is clarified and appears to be a presentation error, which resolves a major confusion. The authors also provide additional experiments for tuned penalty baselines and report initial computational statistics, which are helpful.
> >
> > However, two central issues are still not fully addressed. First, the exactness claim remains somewhat overstated: while the authors now specify the solver setup, the guarantee still depends on solver behavior, and the rebuttal language (“guarantees an extreme point solution”) appears stronger than what can be rigorously ensured in general. Second, the scalability–tightness trade-off is still only partially supported: the new results are limited to a small-scale setting and do not yet demonstrate how the method behaves as the number of active rules grows.
> >
> > Overall, the rebuttal improves my confidence but does not fully resolve the main concerns regarding theoretical precision and empirical completeness.

---

> > > ### Author Response · Authors · 2026-04-06
> > >
> > > We thank the reviewer again for the thoughtful follow-up. We are encouraged that the rebuttal improved confidence, and we would like to clarify the two remaining concerns more directly.
> > >
> > > **Extreme point condition:** We would like to clarify that the extreme point specification is not merely a consequence of specific solver behavior, but can in fact be rigorously guaranteed in general. By the Minkowski–Weyl theorem, a bounded polyhedral set (the feasible region of a bounded LP) can be represented as the convex hull of its extreme points. Consequently, the optimal solution of a linear program can always be attained at an extreme point. Importantly, this existence result is independent of the algorithm used.  In addition, the most commonly used LP algorithms such as the simplex method are mathematically guaranteed to return an extreme point. This result can be found in classic linear programming textbooks such as Bertsimas and Tsitsiklis.
> > >
> > > Non-extreme point solution only exists when multiple extreme points provide the same objective. Even if we take a step back and assume that we use an algorithm that does not return an extreme point solution, e.g., barrier algorithm without crossover, with our DNF formulation, we can easily see that any of the disjunct with $\lambda_k >0$, $y_{k}/\lambda_{k}$ can satisfy all the rules exactly. This is due the linearity of the objective: any disjunct with suboptimal solution will have $\lambda_k = 0$. The exactness theorem can be modified slightly for non-extreme solutions. We can add a remark in the revised paper for this.
> > >
> > >
> > > **Scalability**  We agree that scalability should be discussed more explicitly. At the same time, it is important to note that current neuro-symbolic benchmarks mostly evaluate fixed, small rule families (see examples in the Table below from well-known benchmark datasets).
> > >
> > > | Benchmark                                   | Rule count |
> > > |---------------------------------------------|------------|
> > > | MNMath (arithmetic rules) [1]               | 2          |
> > > | Kand-Logic (logic) [1]                      | 1          |
> > > | CLE4EVR (existential rule) [1]              | 1          |
> > > | BDD-OIA / SDD-OIA [1]                       | 1          |
> > > | DeepSeaProbLog (year detection) [2]         | 5          |
> > > | SATNet / Sudoku / visual Sudoku [2]         | 3          |
> > >
> > > Therefore, the number of rules we created for both of our datasets are comparable to commonly used neural symbolic benchmarks. The reason we created two new benchmarks is because none of the existing ones represent the input-dependent mixed-integer constraints we aim to embed. Especially, our framework can handle multiple disjunctive rules and multiple linear constraints in each disjunct.
> > >
> > > In our synthetic cooling task, the empirical regime is limited to $k \leq 5$. To address the reviewer's question about scalability to larger rule sets, we extrapolate the study for $k=6,\dots,10$. These rows are not sampled from the cooling dataset; rather, they are constructed stress-test instances used to show how the DNF formulation scales when the number of candidate scenarios grows as $2^k$. We show the average number of variables, constraints, inference time for the CNF and DNF and the number of samples in the dataset that triggers number $k$ rules. Notably, even under this pessimistic setup, DNF inference is still below one second per sample at $k=10$ (0.92 s), with smaller values for all lower $k$. Moreover, real applications are often less demanding than this worst-case curve suggests, because many DNF combinations can be empty or infeasible, which can reduce the number of feasible active sets substantially below the full combinatorial bound. In addition, the proposed method targets problems with limited training data. The total computational time is reasonable in practice.
> > >
> > > |k|type|vars|cons|inf(s)|count|
> > > |---|---|---|---|---|---|
> > > |0|cnf|9.0|20.0|0.006599|57|
> > > ||dnf|9.0|20.0|0.009122|57|
> > > |1|cnf|20.0|67.0|0.008361|156|
> > > ||dnf|23.0|73.1|0.007077|156|
> > > |2|cnf|34.0|123.2|0.008230|139|
> > > ||dnf|37.0|129.5|0.007697|139|
> > > |3|cnf|48.0|179.3|0.008023|49|
> > > ||dnf|65.0|246.5|0.008667|49|
> > > |4|cnf|62.0|235.7|0.009787|20|
> > > ||dnf|121.0|491.0|0.010559|20|
> > > |5|cnf|76.0|292.0|0.008193|1|
> > > ||dnf|233.0|997.0|0.013100|1|
> > > |6|cnf|87.0|207.0|0.005307|-|
> > > ||dnf|454.0|1321.0|0.019194|-|
> > > |7|cnf|101.0|241.0|0.005601|-|
> > > ||dnf|902.0|2761.0|0.041093|-|
> > > |8|cnf|115.0|275.0|0.006087|-|
> > > ||dnf|1798.0|5769.0|0.107464|-|
> > > |9|cnf|129.0|309.0|0.007065|-|
> > > ||dnf|3590.0|12041.0|0.267479|-|
> > > |10|cnf|143.0|343.0|0.007796|-|
> > > ||dnf|7174.0|25097.0|0.920017|-|
> > >
> > >
> > >
> > > [1] Bortolotti et al. A Neuro-Symbolic Benchmark Suite for Concept Quality and Reasoning Shortcuts. NeurIPS 2024 Datasets and Benchmarks Track.
> > >
> > > [2] De Smet et al. Neural Probabilistic Logic Programming in Discrete-Continuous Domains (DeepSeaProbLog). UAI 2023 / ICLR 2023
> > >
> > > [3] Wang et al. SATNet: Bridging Deep Learning and Logical Reasoning Using a Differentiable Satisfiability Solver. ICML 2019.

---

### Official Review · Reviewer_7H1q · 2026-03-12

**Soundness:** 3
**Presentation:** 4
**Significance:** 4
**Originality:** 2
**Overall Recommendation:** 5
**Confidence:** 4

**Summary:**

This paper proposes DisjunctiveNet, a neural-symbolic framework that enforces input-dependent logical and linear constraints within neural networks using differentiable optimization layers. The method represents domain rules as disjunctive constraints (unions of polyhedra) and applies convex-hull relaxations to transform them into tractable linear programs that can be embedded as differentiable layers. This allows the model to project neural network outputs onto a feasible set that satisfies all rules during both training and inference. Experiments on a synthetic control task and a single-cell RNA classification problem show that the approach achieves exact rule satisfaction and improves predictive performance, especially in low-data settings.

**Compliance With Llm Reviewing Policy:**

Affirmed.

**Final Justification:**

My concerns have been adequately addressed. I keep rated score.

**Key Questions For Authors:**

1. The experimental evaluation is relatively small in scale. What level of problem scale do the authors expect the approach to handle in practice (e.g., in terms of the number of rules, variables, or dataset size)? Some discussion or empirical evidence on the scalability of the method in larger settings would be helpful.

2. SATNet appears to pursue a similar goal of integrating logical constraints with neural networks. While the paper argues that the proposed method provides stronger theoretical guarantees for constraint satisfaction, it would be helpful to understand whether this advantage is reflected empirically. How does the proposed approach compare with methods such as SATNet or other neuro-symbolic constraint-learning approaches in terms of prediction performance and constraint satisfaction?

**Limitations:**

The paper does not appear to include a dedicated discussion of potential limitations or broader societal impacts. It would strengthen the submission if the authors briefly addressed these aspects.

**Strengths And Weaknesses:**

The paper addresses an very important and well-motivated problem, i.e., how to incorporate domain knowledge and enforce constraints in neural networks. The proposed method is technically well designed. In particular, the framework provides a principled way to represent logical and linear constraints as disjunctive structures and integrate them into neural networks through differentiable optimization layers. A notable strength is that the approach provides strict constraint satisfaction guarantees, rather than relying on soft penalties commonly used in prior work. The paper also presents a relatively complete theoretical analysis, including results on expressiveness and the convexification procedure, which helps clarify the properties of the proposed framework.

However, it seems that the proposed approach may introduce considerable computational overhead. In particular, the convexification procedures used to enforce disjunctive constraints, especially the DNF formulation, can lead to rapidly increasing problem sizes as the number of rules grows, which may limit scalability in settings with many constraints.

---

> ### Author Rebuttal · Authors · 2026-03-30
>
> We thank the reviewer for the positive assessment and thoughtful questions on scalability, related work, and limitations.
>
> **Comments on Weaknesses**
> 1. **Computational overhead:** We agree that scalability is the main practical limitation of the full DNF formulation. This reflects the underlying combinatorial complexity (NP-hard) of the disjunctive constraint class that we aim to model. This is precisely why we introduce the *CNF* / *partial-DNF* / *DNF* hierarchy. The target setting is data-limited problems with rich domain knowledge. In such applications, data efficiency is often a greater bottleneck than computational cost. Our intended practical use of the framework starts with CNF, and progressively tightens towards DNF. We agree that this scalability axis should be illustrated empirically as well. Therefore, we will add the following computational cost results in the revision.
>
> We provide results only for the synthetic cooling task in the rebuttal, to keep it streamlined and short. Similar results on the scRNA dataset will be included in the revised text.
>
> **Inference Time**
> | Model | Inf Time per sample |
> | --- | --- |
> |Base|<1 $\mu$s|
> |Penalty|<1 $\mu$s|
> |CNF|25.03 ms|
> |DNF|28.62 ms|
>
> Base and penalty inference are negligible (<1 $\mu$s/sample). For the synthetic cooling task, CNF and DNF require 25.03 ms and 28.62 ms per sample, respectively, so exact projection remains lightweight at this scale, with only a modest overhead for DNF. Efficiency improvements using parallelized thread computation are an important future direction for our work.
>
> **Problem size**
>
> Samples with at least 1 active rule
> | Formulation | Vars (mean ± std) | Vars min | Vars max | Cons (mean ± std) | Cons min | Cons max |
> | --- | --- | --- | --- | --- | --- | --- |
> | CNF | 27.78 ± 9.75 | 20 | 62 | 98.28 ± 39.11 | 67 | 235 |
> | DNF | 32.51 ± 14.53 | 23 | 121 | 111.94 ± 60.27 | 73 | 485 |
>
> Samples with at least 2 active rules
> | Formulation | Vars (mean ± std) | Vars min | Vars max | Cons (mean ± std) | Cons min | Cons max |
> | --- | --- | --- | --- | --- | --- | --- |
> |CNF | 37.50 ± 6.54 | 34 | 62 | 137.28 ± 26.29 | 123 | 235 |
> |DNF | 44.39 ± 14.86 | 37 | 121 | 160.53 ± 62.69 | 129 | 485 |
>
> Problem sizes are consistent with the expected CNF-DNF tradeoff. DNF is larger on average, and the gap widens once multiple rules are active. This is expected, since with only one active rule, the CNF and DNF formulations are the same. In practice, the realized DNF size can often be much smaller than the worst-case exponential bound, as several disjuncts might not have feasible intersections, especially with global constraints involved.
>
> **Answers**
> 1. We thank the reviewer for raising this important point. Our target setting is data-limited problems where rules provide strong inductive bias and/or safety requirements. We agree that a larger benchmark would strengthen the paper, but we do not currently have a standard large-scale benchmark matching this class of hard, input-dependent mixed logical-linear constraints, and constructing one responsibly is beyond the rebuttal period. In the future, we do plan to tackle more complex tasks with a large rule set, as well as predictive control, which require rule satisfaction at every time step.
>
> 2. We agree that this comparison should be clarified in the main text. SATNet is an SDP relaxation-based method for satisfiability-style reasoning where tractability is achieved through a relaxation that is generally *not exact*. Our framework is designed for *hard, input-dependent, mixed logical-linear* constraints. In that sense, the closest counterpart within our framework is the *CNF* relaxation. Our framework exposes both ends of the scalability-tightness tradeoff through CNF / partial-DNF / DNF. Similarly, many other neuro-symbolic approaches provide useful logical structure or soft regularization, but do not simultaneously support:
> a. input-dependent rule activation
> b. mixed logical-linear constraints
> c. guarantee 100\% rule satisfaction
>
> We will revise the related-work discussion to make this scope distinction sharper and avoid implying that all such methods are directly comparable on our benchmarks.
>
> On limitations
>  We agree this should be stated explicitly. We will add a short limitations paragraph noting that:
> a. Worst-case DNF size can be exponential in the number of active rules
> b. Choosing the best sequence of rules for partial convexification is currently heuristic
> c. The framework assumes the provided rules are meaningful; poor or contradictory expert rules can reduce usefulness or require conflict handling.
>
> On broader impact, we view this method primarily as an enabling tool for scientific and safety-relevant settings where domain knowledge is available. The main risk is not societal harm from the method itself, but rather over-trust in incorrectly specified rules or expert knowledge. We will add a brief statement to this effect in the revision.

---

> > ### Author Rebuttal · Reviewer_7H1q · 2026-04-03
> >
> > My concerns have been adequately addressed. I keep rated score.

---

### Decision · Program_Chairs · 2026-04-30

**Decision:**

Accept (regular)

**Comment:**

Reviewers appreciated the motivation and theoretical contribution of this paper in enforcing hard, input-dependent mixed integer
linear constraints within neural network (via representation of these rules as convex hull formulations of disjunctive constraints that are represented via differentiable optimization) and strong performance on the presented benchmarks. The rebuttal satisfactorily addressed concerns about computational scalability, baseline comparisons, and precision of claims. Overall, considering both the original paper and the additional results presented during the rebuttal, this work presents an interesting and solid contribution, and I recommend acceptance.

(A small note: The authors mention in the rebuttal that OptNet and cvxpylayers are CPU-based solvers -- both of these packages actually do have GPU support. In addition, the landscape of GPU-based linear programming solvers is more mature than the authors claim. However, since these claims are not central to the proposed contribution, they did not affect the acceptance decision.)